# Explosive eruption style modulates volcanic electrification signals
Caron E. J. Vossen [1] ✉, Corrado Cimarelli [1], Luca D'Auria[2,3], Valeria Cigala[1], Ulrich Kueppers [1], José Barrancos[2,3] & Alec J. Bennett [4,5]

Volcanic lightning detection has proven useful to volcano monitoring by providing information on eruption onset, source parameters, and ash cloud directions. However, little is known about the influence of changing eruptive styles on the generation of charge and electrical discharges inside the eruption column. The 2021 Tajogaite eruption (La Palma, Canary Islands) provided the rare opportunity to monitor variations in electrical activity continuously over several weeks using an electrostatic lightning detector. Here we show that throughout the eruption, silicate particle charging is the main electrification mechanism. Moreover, we find that the type of electrical activity is closely linked to the explosive eruption style. Fluctuations in the electrical discharge rates are likely controlled by variations in the mass eruption rate and/or changes in the eruption style. These findings hold promise for obtaining near real-time information on the dynamic evolution of explosive volcanic activity through electrostatic monitoring in the future.

On 19 September 2021 at 14:10 UTC, an eruption started from a fissure on the Western flank of the Cumbre Vieja volcanic ridge on La Palma (Canary Islands), 50 years after the last eruption in 1971[1]. For almost three months, several vents along a NW-SE aligned fissure (Fig. 1) erupted lava and tephra of basanite to tephrite composition[2,3]. The eruption resulted in a 12 km² compound Aā and Pāhoehoe lava flow field[4], which destroyed entire villages, infrastructure, and plantations, and formed a 187 m tall scoria cone (1071.2 m above sea level) later named Volcán de Tajogaite[5]. The eruption ended on 13 December 2021. Throughout the eruption, the explosive activity varied on the order of hours to days, ranging between mild to strong ash emissions, gas jetting, Strombolian activity, and lava fountaining[6,7]. On many occasions, volcanic lightning was observed. Note that there is some debate about how to classify the different eruption styles during this eruption due to the high variability in activity without clear boundaries between one activity and another[8]. The nomenclature used in this study is following Romero et al.[7].

Volcanic lightning is frequently observed during ash-rich explosive eruptions. It is interpreted as a result of electrification and charge separation in the eruption column[9]. For plumes that do not reach atmospheric freezing levels, the dominant charging mechanism is silicate particle charging, through fracturing of[10,11] and/or collision of particles[12–15]. If sufficiently high plumes are generated, ice nucleation can further enhance the plume electrification during more evolved stages of the eruption[16,17]. Generally,

volcanic ash becomes an effective catalyst for ice nucleation below −20 °C[18,19]. The majority of volcanic lightning studies were focused on major eruptions using data from global lightning networks, such as Vaisala GLD360 Global Lightning Detection[17,20,21], Earth Networks Total Lightning Network[16] and the Global Volcanic Lightning Monitor of the World Wide Lightning Location Network (WWLLN)[20,22]. However, the 2021 Tajogaite eruption passed under the radar of these networks due to the lack of nearby sensors as well as relatively low-amplitude volcanic lightning (in comparison to major eruptions), demonstrating that local electrical detectors are required[23,24]. This eruption provided the opportunity to continuously detect the electrical activity throughout transitioning eruption styles and intensities, which would vary on the order of hours to days. With the aim to link different electrical signals to varying explosive activity, electrostatic data from a Biral Thunderstorm Detector (BTD) was combined with thermal videography, visual imaging, standard atmospheric measurements, and volcanic tremor measurements.

Here we demonstrate that electrical discharges were generated almost continuously throughout the time of monitoring and that silicate particle charging was the main driver for plume electrification during this eruption. In addition, we find that transitions in the explosive activity, sometimes accompanied by sudden shifts in the seismic tremor amplitude, can be distinguished based on distinct changes in the electrical signature.

[1]Department of Earth and Environmental Sciences, Ludwig-Maximilians-Universität München, Munich, Germany. [2]Instituto Volcanológico de Canarias (INVOL-CAN), 38320 La Laguna, Tenerife, Canary Islands, Spain. [3]Instituto Tecnológico y de Energías Renovables (ITER). Polígono Industrial de Granadilla, s/n 38600 - Granadilla de Abona, Santa Cruz de Tenerife, Spain. [4]Bristol Industrial and Research Associates Ltd (Biral), Unit 8 Harbour Road Trading Estate, Portishead, Bristol BS20 7BL, UK. [5]Department of Electronic and Electrical Engineering, University of Bath, Bath, UK. ✉e-mail: caron.vossen@min.uni-muenchen.de

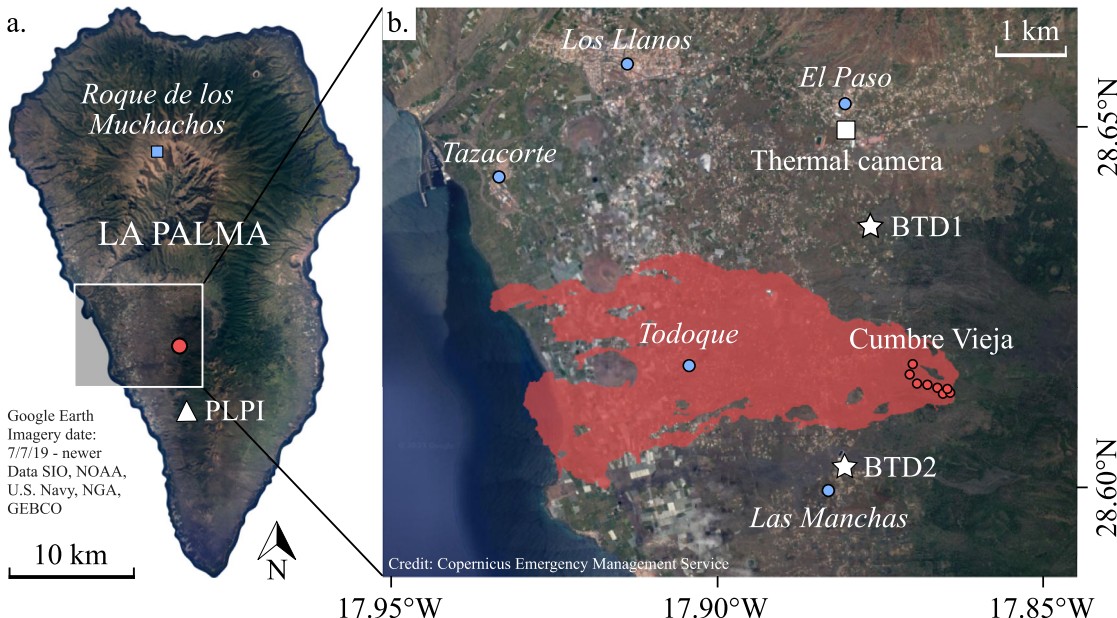

**Fig. 1 | Map of La Palma with the locations of Cumbre Vieja volcanic ridge and the instruments.** Google Earth satellite images (Imagery date: 7/7/2019 – newer. Data SIO, NOAA, U.S. Navy, NGA, GEBCO. http://www.earth.google.com [24 October 2023]) **a** La Palma, Canary Islands (Spain), showing the location of Volcán de Tajogaite (red circle), seismic station PLPI (white triangle) and Roque de los Muchachos observatory (blue square). **b** The 2021 Tajogaite lava flow field (red shaded area; data from the European agency Copernicus Emergency Management Service, https://emergency.copernicus.eu/mapping/list-of-components/EMSR546 [24 October 2023]) is shown together with the location of the Biral Thunderstorm Detector (installed from 11-26 October at location BTD1 and relocated to location BTD2 on 27 October, white stars), the thermal camera (white square), the active vents (red circles) and nearby villages (blue circles).

## Results

### Electrical activity, seismic tremor, and plume height

Details of the monitoring setup, data collection[25], and analysis are provided in the Methods section. Plume height data was obtained from two different organisations, the Toulouse Volcanic Ash Advisory Center (VAAC) and Plan de Emergencias Volcánicas de Canarias (PEVOLCA). These two datasets are compared with the time series of atmospheric temperature at the 0 °C, −10 °C and -20 °C isotherms (Fig. 2a), the average and maximum value of the absolute voltage (V) per hour (Fig. 2b) and the electrical discharge rate (discharges per hour) as detected by the BTD and identified by the volcanic lightning detection algorithm during the observation period (Fig. 2c). It is important to take into account that the detection efficiency differs between the two BTD locations (BTD1 and BTD2) as the electric field decreases with the distance cubed at frequencies <100 Hz[26,27]. Besides the known distance between the BTD and the active vents, also the generally unknown and constantly changing height of the electrical discharges within the plume and the height and movement of the charged plume itself with respect to the BTD affect the amplitude of the electric field. In general, BTD2 had a higher detection efficiency due to its closer location to the active vents. This affects the electrical parameters calculated from the measurements.

The results show that volcanic ash primarily stayed below atmospheric freezing levels throughout the eruption, with an average maximum height of ~2968 m above sea level (a.s.l.) based on both datasets. On 28 September and 1 October, Toulouse VAAC and PEVOLCA did report the presence of volcanic ash at altitudes above the −10 °C isotherm, respectively, but the BTD had not yet been deployed at this time. Between 18:00 and 03:00 UTC on the night of 13-14 December, volcanic ash was observed at its highest level (almost 8 km a.s.l.), exceeding the −20 °C isotherm, as a result of intense ash-rich lava fountaining. A relatively small increase in the electrical discharge rate was detected in response to this activity. This explosive phase stopped around 21:30 UTC on 13 December marking the end of the 2021 Tajogaite eruption. Nonetheless, ash remained suspended at high altitudes for several hours longer. Toulouse VAAC did not detect any volcanic ash after 12:46 UTC on 15 December.

Electrical discharges were detected almost continuously, indicating that the eruption was very electrically active. In general, there is no clear correlation between the plume height, the electrical discharge rate, and the average and maximum voltage measured by the primary antenna (Fig. 2a–c). There are short periods however, e.g. 29 September – 4 November, where fluctuations in the Toulouse VAAC plume height dataset are positively correlated to the overall changes in the electrical discharge rate. In contrast, between 1 and 3 December a strong increase in the electrical discharge rate was recorded while the eruption column height decreased more than 1.5 km, indicating an anticorrelation. The BTD recorded an electrical discharge rate >5000 discharges per hour on various occasions throughout the time of monitoring. These discharge rates were detected during different eruption styles, including strong ash emissions, ash-rich lava fountaining and intense Strombolian activity, as was observed both during a field campaign in early November 2021 and through videos posted by Instituto Volcanológico de Canarias on X (Twitter). The normalized discharge rate, which is the electrical discharge rate times the normalized maximum voltage measured per hour by the primary antenna (see Methods section), demonstrates that there is no relationship between these two parameters (Fig. 2b, c). The average and maximum voltage fluctuate strongly throughout the eruption, with the primary antenna repeatedly reaching the saturation level of 0.785 V (Fig. 2b).

Both the Very Long Period (VLP, 0.4–0.6 Hz) and the Long Period (LP, 1–5 Hz) tremor amplitude varied throughout the eruption (Fig. 2d). The sharp decrease in amplitude on 27 September coincides with a temporary cease in the eruption shortly after the cone collapses on 25 September[7,8]. Similarly, the sudden changes in amplitude on 12 and 13 December are related to a short phase of quiescence followed by the highly explosive phase on the evening of 13 December. Since the volcanic tremor amplitude is highly variable, the temporal variation of the LP and VLP components was additionally analyzed using the Principal Component Analysis (PCA). The visual comparison of signals in Fig. 2 demonstrates that the first principal component (PC1) is mostly related to changes in the absolute tremor amplitude (Fig. 2e),

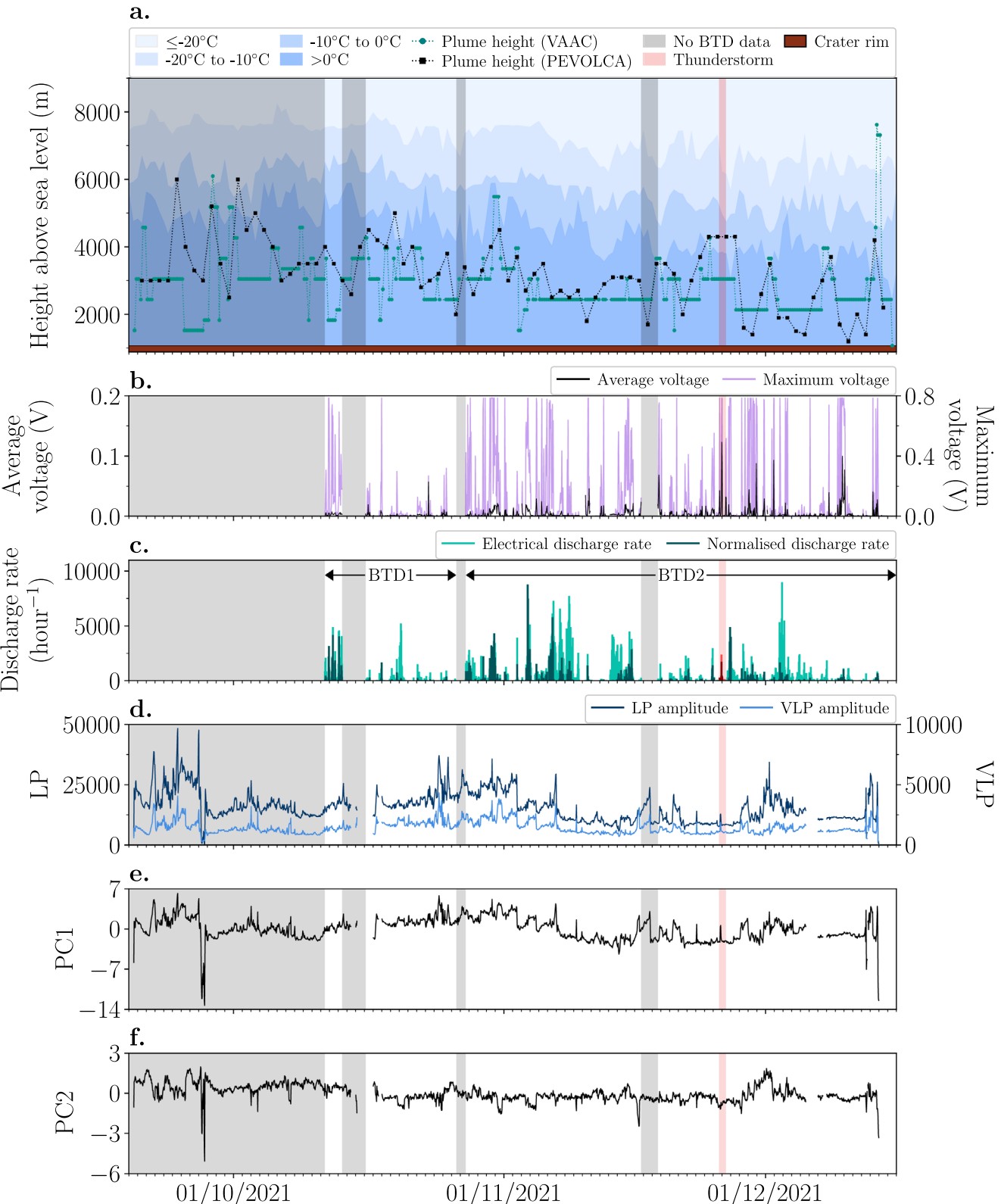

**Fig. 2 | Electrical, seismic tremor, and plume height measurements throughout the 2021 Tajogaite eruption. a** Volcanic plume heights as reported by the Toulouse Volcanic Ash Advisory Center (VAAC, black squares) and the Plan de Emergencias Volcánicas de Canarias (PEVOLCA, green circles) are compared to different temperature regions (shades of blue) as a function of height (m) above sea level. The elevation of the crater rim varied between 884.2 and 1071.2 m above sea level due to cycles of growth and collapse[5]. The red shaded area marks a local thunderstorm on 25 and 26 November. The grey shaded areas denote the periods for which no BTD data is available. **b** The one-hour average (black line, left y-axis) and maximum (purple line, right y-axis) of the absolute voltage measured by the primary antenna of the BTD. **c** The electrical discharge rate (light green vertical bars) and the normalised discharge rate (dark green vertical bars) per hour. The normalised discharge rate is the electrical discharge rate times the normalised maximum voltage measured per hour by the primary antenna. The light and dark red vertical bars show the electrical and normalised discharge rate, respectively, measured during a nearby thunderstorm. The arrows indicate the time of monitoring at each location (BTD1 and BTD2). Median calculated over one hour for **d** LP (dark blue line, left y-axis) and VLP (light blue line, right y-axis) amplitudes; **e** First principal component (PC1); **f** Second principal component (PC2).

while the second principal component (PC2) is mainly dependent on their ratio. Based on findings from Bonadonna et al. [6], this suggests that PC1 is related to the intensity of the explosive activity, whereas PC2 reflects the changes in the volcanic tremor source mechanism, which is in turn connected to the type of volcanic activity. A more detailed explanation is provided in the Methods section. Also for the volcanic tremor signals and the result of the PCA applies that there is no evident correlation with the plume height and the electrical parameters.

## Types of electrical activity

A variety of electrical signals was detected throughout the course of the eruption, which would change frequently on the order of hours. In general, six main types of electrical signatures were observed, hereafter referred to as types 1-6:

1. Individual high-amplitude electrical discharges (typically >0.01 V, Fig. 3a).
2. Minutes-long "bursts" of quasi-continuous low-amplitude electrical activity, generally ranging between 0.001–0.01 V (Fig. 3b).
3. Seconds-long (~2–45 s) "bursts" of quasi-continuous low-amplitude electrical activity, commonly below 0.005 V (Fig. 3c).
4. Faint electrical discharges with a very low amplitude (<0.002 V) recorded by the primary antenna (Fig. 3d). The sensitivity of the secondary antenna is too low to detect any electrical activity. For this reason, these discharges remain mostly undetected by the detection algorithm.
5. Movement of charge, visible as slow variations (~1–10 s) in the electrostatic signal. Simultaneous electrical discharges are superimposed and can still be detected by the detection algorithm (Fig. 3e).
6. Ash fall on top of the sensor can be discriminated from electrical discharges as these impact transients produce electrical signals of opposite polarities at the two antennas, resulting in a negative covariance (Fig. 3f). Electrical discharges can still be detected during ash fall, as is shown by the positive covariance at 10:27:56.6 UTC in Fig. 3f.

The ranges in measured voltage for signal types 1–4 are based on the detections of BTD2, which are generally of higher amplitude compared to BTD1 due to its closer location to the active vents. Combinations of different types were frequently observed. In particular, the individual high-amplitude discharges (signal type 1) were often accompanied by movements of charge (signal type 5). Also, mixtures of seconds- and minutes-long bursts of quasi-continuous low-amplitude electrical activity (signal types 2 and 3) were detected. Moreover, short periods of ash fall were regularly detected during the varying types of electrical activity.

## Linking electrical activity to explosive eruption styles

During a field campaign in November 2021, changes in the explosive activity style and intensity were observed every few hours. Here, we compare in detail the electrical signals and electrical discharge rate detected at BTD2 to visual and thermal images as well as the volcanic tremor time series and corresponding PCA for the evening of 3 November and the entire day of 4 November (Fig. 4). Close-up plots of the different electrical signatures are provided in Supplementary Fig. 4 of the Supplementary Results.

Between 21:00 UTC on 3 November and 00:03 UTC on 4 November, BTD2 recorded individual high-amplitude electrical discharges (signal type 1), which frequently exceeded a measured voltage of 0.1 V (Fig. 4b, c and Supplementary Fig. 4b). This electrical activity was recorded during ash-rich lava fountaining from the main active vent (Figs. 4a-1), which produced a 3 km a.s.l. tall ash plume (Toulouse VAAC data; Fig. 2a). The thermal videography showed an initial diameter of 300-500 m of the eruption column with relatively dense, turbulent eddies (Supplementary Fig. 5a). At this time, many flashes of volcanic lightning were detected (Fig. 4c). For short periods of time, pyroclasts were ejected from a second vent as well (Supplementary Fig. 5a).

After 00:03 UTC, the electrical discharges occurred less frequently and gradually decreased in magnitude (<0.1 V, Fig. 4b, c and Supplementary Fig. 4c). This progressive decline in the electrical activity corresponds to a decrease in plume height as reported by Toulouse VAAC (Fig. 2a), indicating a waning of the explosive activity. These changes are accompanied by a high variability in the tremor amplitudes and the value of PC1, while the value of PC2 shows a descending trend (Fig. 2d, f). Due to the lower magnitude of the electrical discharges, the movement of charge (signal type 5) produced by the moving electrified ash plume relative to the BTD, becomes visible in the electrical data (Supplementary Fig. 4c).

From approximately 03:00 UTC on, the lava fountaining phase stopped and mild ash emission producing a small and well-defined ash plume (basal diameter <100 m) was observed (Figs. 4a-2). This was accompanied by very faint (<0.001 V) electrical discharges (Supplementary Fig. 4d) that predominantly remained undetected by the algorithm (signal type 4). Consequently, the electrical discharge rate is very low during this period (Fig. 4c).

The explosive activity subsequently changed into a phase of gas jetting with minor amounts of particles being ejected (Figs. 4a-3). This change was preceded by a temporary increase in both the tremor amplitudes and the value of PC1 and a temporary decrease in the value of PC2. At times, shock waves were visible, indicating gas expansion velocities above the speed of sound. Between 13:09-14:16 UTC, BTD2 detected ~3-10 seconds long bursts of quasi-continuous electrical activity with a measured voltage commonly below 0.001 V (signal type 3; Fig. 4c and Supplementary Fig. 4e). From the thermal images can be deduced that cooling of the pyroclasts is of minor importance in the first 300 m of vertical transport and that the pyroclasts continued to rise up to 1000 m above the crater rim (a.c.r.). Note that the higher amplitude signals recorded between 13:45-13:50 UTC are an artefact of carrying out maintenance of the BTD. From 14:28 UTC on, ash and lapilli started falling at the location of BTD2, occasionally resulting in a signal of negative covariance. Between 15:10-15:34 UTC, the quasi-continuous electrical bursts increased in duration (up to ~45 seconds) and generally had a slightly higher measured voltage (<0.002 V, Fig. 4c and Supplementary Fig. 4f). Simultaneously, an increase in the VLP amplitude was recorded, resulting in an increase in the value of PC1 and a decrease in the value of PC2 (Fig. 4d–f). During this time, the thermal images show an increment in the intensity of the gas jetting, evident from the increase in the total amount of hot material that is ejected up to greater heights (up to 700 m a.c.r.).

From 15:46 UTC onward, ash fall increased at BTD2 (along with a few minutes of light rain), based on the prevalent detection of electrical signals of opposite polarity at the two antennas (signal type 6, Supplementary Fig. 4g) as well as direct observations. This coincided with a change in the activity from gas jetting to ash emissions at two vents simultaneously. During this period, both the tremor amplitudes and the value of PC1 gradually decreased, while the value of PC2 gradually increased (Fig. 4d–f). Around 16:57:43 UTC, the explosive activity increased in intensity and changed rapidly from mild to very strong ash emission, as is manifested in the rise diagram by the increase in the average height and temperature of the eruption column (Figs. 4a-4 and 5). This sudden change in activity was accompanied by a sharp decrease in both tremor amplitudes and the value of PC1, as well as a distinct increase in the value of PC2 (Fig. 4d–f). Shortly after at 16:57:50 UTC, individual high-amplitude electrical discharges (>0.1 V, signal type 1) were detected (Fig. 4b, c and Supplementary Fig. 4h), which lasted for almost 1.5 hours. The ash plume reached a height of approximately 3 km a.s.l. according to the Toulouse VAAC (Fig. 2).

Around 18:22 UTC, the electrical discharge rate and the maximum value of the absolute voltage decreased (Fig. 4c). This change was accompanied by a brief increase in the seismic tremor amplitudes and the value of PC1, as well as a temporary decrease in the value of PC2 (Fig. 4d–f). During this time, the thermal data showed that the vent area was obscured by clouds most of the time, and therefore no information is available on the explosive activity. It can be speculated, however, based on the measurements that the activity was starting to change at this point. Around 18:46 UTC, once the

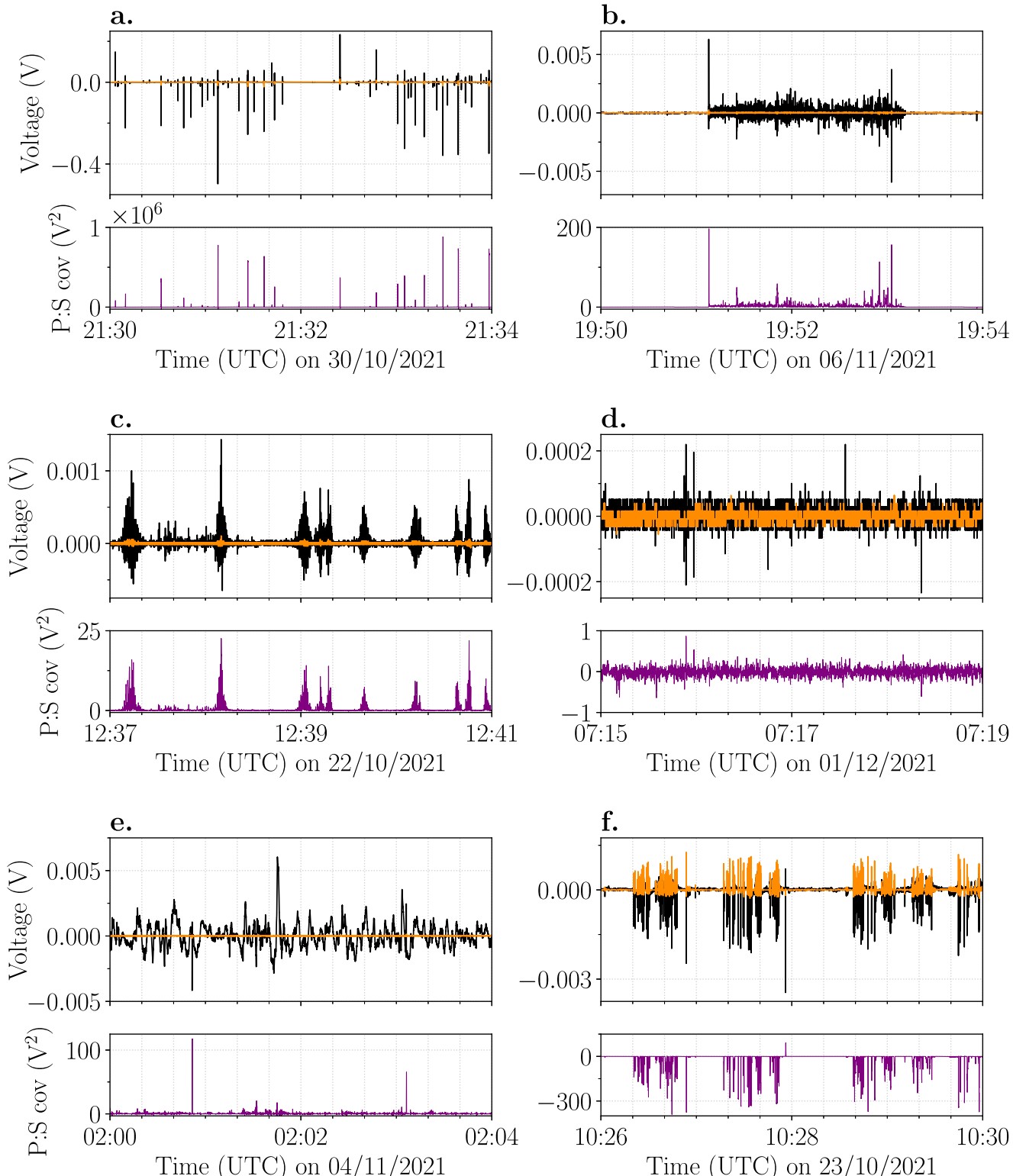

**Fig. 3 | Different electrical signals recorded during the 2021 Tajogaite eruption.** Voltage (V) measured by the primary (black line) and secondary (orange line) antennas of the BTD with the corresponding covariance ($V^2$) between the primary and secondary signals shown below (purple line). Note the different scales on the y-axis for each panel. **a** Individual high-amplitude electrical discharges; **b** Minutes-long bursts of quasi-continuous low-amplitude electrical activity; **c** Seconds-long bursts of quasi-continuous low-amplitude electrical activity; **d** Faint electrical discharges that generally remain unidentified by the detection algorithm; **e** Movement of charge, indicated by the slow-varying electrostatic signal. **f** Ash falling on top of the sensor, evidenced by the negative covariance. A single electrical discharge was detected at 10:27:56.6 UTC, indicated by the corresponding positive covariance.

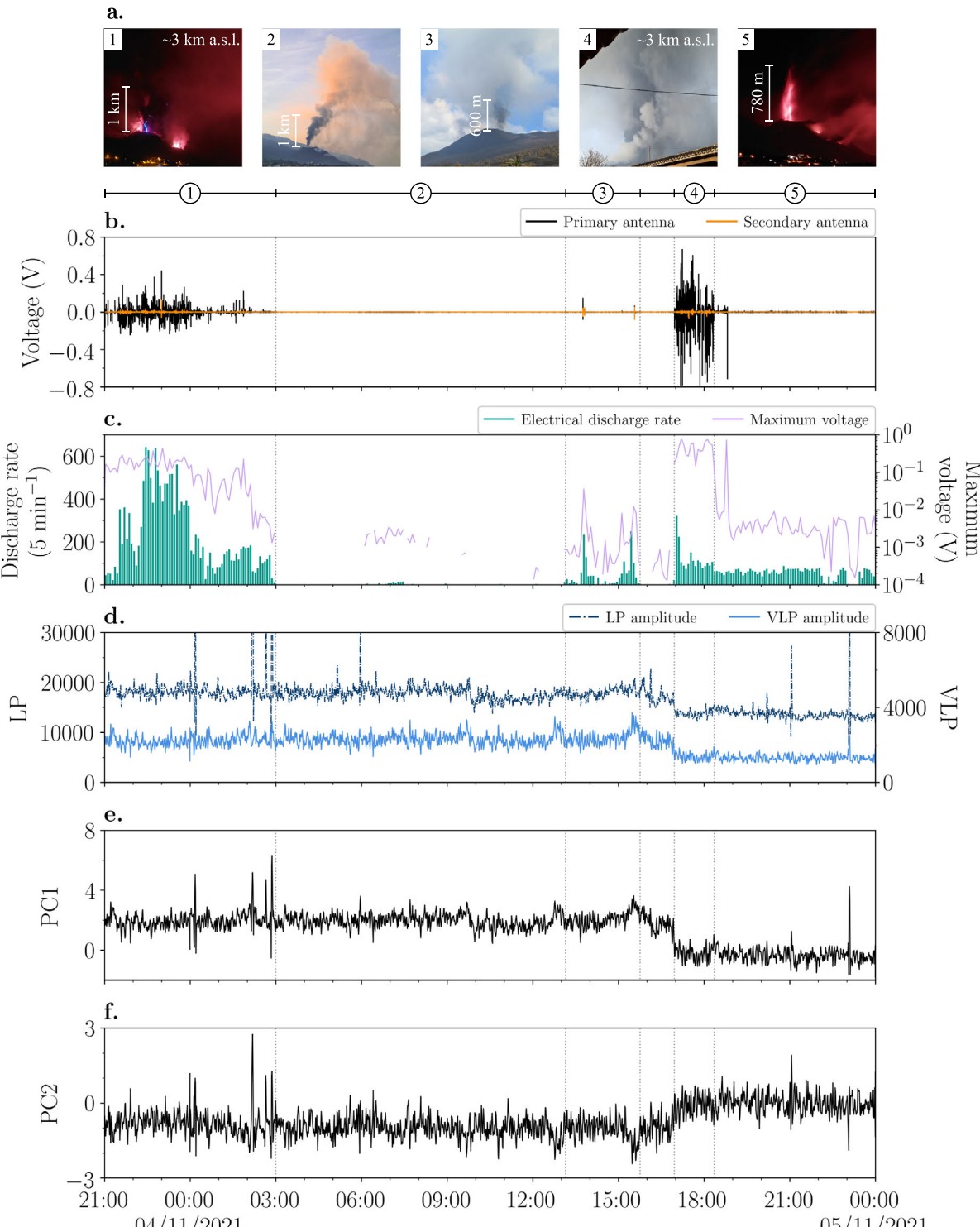

**Fig. 4 | Explosive activity on 3 and 4 November and the corresponding electrical and volcanic tremor signals. a** Pictures of different explosive eruption styles on 3 and 4 November: (1) Ash-rich lava fountaining at 23:30:53 UTC on 3 November, producing a plume of approximately 3 km a.s.l. and a flash of volcanic lightning. Courtesy of Francisco Cáceres Acevedo. Reproduced with permission of the copyright holder; (2) Mild ash emissions at 07:44 UTC on 4 November, producing a small ash plume; (3) Gas jetting at 13:11 UTC on 4 November; (4) Strong ash emissions at 17:08 UTC on 4 November, producing a plume of approximately 3 km a.s.l.; (5) Lava fountaining without the generation of a large ash plume in contrast to (1). Picture was taken at 21:51 UTC on 4 November. Courtesy of Francisco Cáceres Acevedo. Reproduced with permission of the copyright holder. **b–f** Measurements taken between 21:00 UTC on 3 November until 00:00 UTC on 5 November. Numbered

pictures in panel **a** show prevalent activity of the periods 1-5 reported in panels **b-f**. **b** Voltage (V) measured by the primary (black line) and secondary (orange line) antennas of BTD2. Close-up plots of the different electrical signatures are shown in Supplementary Fig. 4. **c** Electrical discharge rate (discharges per 5 minutes) as green vertical bars (left y-axis) and the maximum value of the absolute voltage measured by the primary antenna per 5 minutes in purple (right y-axis with log-scale). **d** Seismic tremor amplitude, showing the LP amplitude (dash-dotted line) on the left y-axis and the VLP amplitude (solid line) on the right y-axis. The peaks were produced by earthquakes. **e** First principal component (PC1), which is mostly related to changes in the absolute tremor amplitude. **f)** Second principal component (PC2), which mainly reflects changes in the volcanic tremor source mechanism.

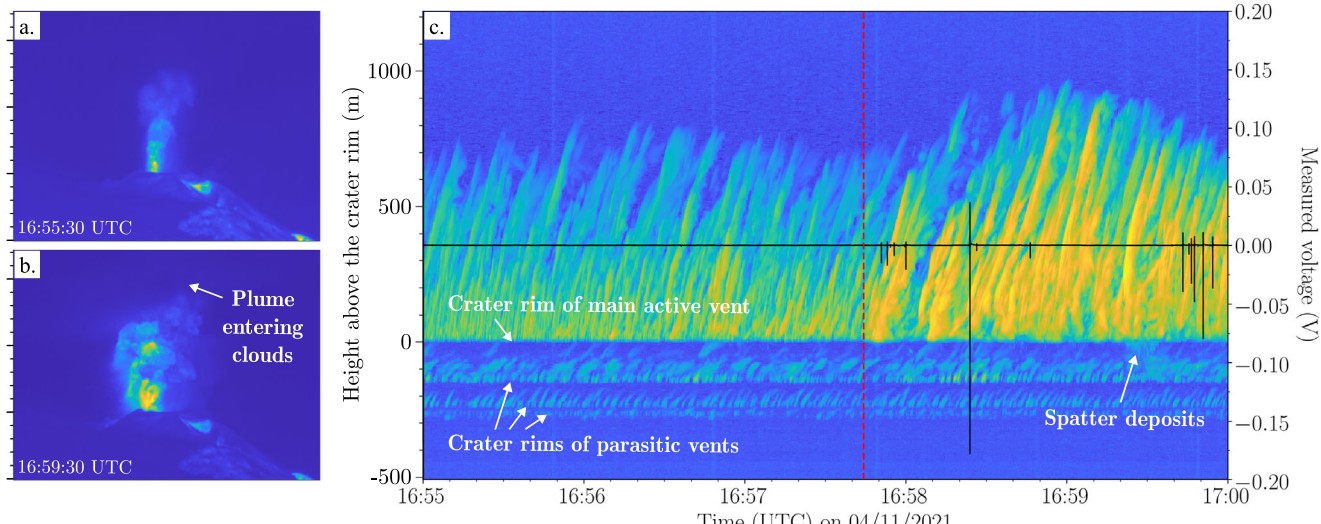

**Fig. 5 | Thermal infrared imaging of changing explosive eruption style.** Single thermal frame of **a** mild ash emissions at 16:55:30 UTC and **b** strong ash emissions at 16:59:30 UTC entering the clouds approximately 900 metres above the crater rim (see Figs. 4a-5). The y-axes correspond to the y-scale of panel **c**. **c** Thermal infrared rise diagram showing the evolution of the maximum temperature (blue is cold and yellow is hot) as a function of time and height (left y-axis). The crater rim of the main active vent is at a height of 0 m in the diagram. At lower heights, currently inactive vents can be seen as thermal anomalies due to ongoing passive degassing. The increase in magnitude of the explosive activity at 16:57:43 UTC (vertical red dashed line) is closely followed by the onset of individual high-amplitude electrical discharges (signal type 1) as recorded by the primary antenna of BTD2 (black line, right y-axis).

clouds had cleared away, the thermal images showed that the explosive activity had indeed changed to a lava-fountaining phase again (Figs. 4a-5), dominated by the ejection of incandescent bombs. In general, the lava fountains reached heights of 500–1000 metres above the crater rim, although individual bombs were occasionally ejected to greater heights. In contrast to the lava fountaining phase on the evening of 3 November, less ash was emitted. As a result, the eruption column only had an initial diameter of 100–200 m and dense, turbulent eddies remained absent (Supplementary Fig. 5b). This change in explosive activity coincided with a change in electrical activity from individual high-amplitude electrical discharges (signal type 1) to predominantly minutes-long bursts with measured voltages generally between 0.001–0.01 V (signal type 2, Fig. 4c and Supplementary Fig. 4i), sometimes interrupted by bursts of shorter duration but similar measured voltage. However, contrary to the signal shown in Fig. 3b, c, these signals did not have only positive covariance values, which is one of the criteria of the volcanic lightning detection algorithm (see Methods section and Vossen et al.[23] for more information). For this case, the negative covariance could result from the low sensitivity of the secondary antenna, or it could be caused by an ash-induced change/lag in the capacitance of one of the antennas due to previous ash fall deposited on the sensor. This does not affect the detection of the electrical discharges by the BTD but does result in an underestimate of the electrical discharge rate as the detection algorithm disregards signals with a negative covariance (Fig. 4c). During this lava fountaining phase, the tremor amplitudes and the values of PC1 and PC2 remained relatively constant with comparison to the previous explosive phase.

Strombolian activity was not observed on 3 and 4 November but was observed for several hours on the evening of 6 November. Compared to the phases of lava fountaining, the incandescent bombs were ejected to lower heights, generally less than 500 metres above the crater rim (Fig. 6). In addition, a partially opaque ash plume was produced. This type of activity predominantly produced a mixture of seconds-long and minutes-long bursts of electrical activity with measured voltages dominantly between 0.001–0.0025 V (signal types 2 and 3, Fig. 6d), although at times single electrical discharges of similar amplitude occurred in between bursts. Comparing three sequences of thermal frames, corresponding to different electrical signals demonstrates that there are small differences in the explosive activity on short timescales, especially in the size of the eruption column. Very minor Strombolian activity, reaching heights up to 200 m a.c.r. (Fig. 6a), produced little to no electrical activity. A burst of quasi-continuous electrical discharges lasting ~2.5 minutes was detected during a period where pyroclasts were ejected up to 400 m a.c.r. (Fig. 6b), while a shorter burst lasting approximately 20 seconds was generated during intermediate Strombolian activity reaching heights of 100-300 m a.c.r. (Fig. 6c). Nonetheless, no obvious correlation between the electrical signals and the rise diagram can be observed that could explain the variation in the duration of the bursts (Fig. 6d). Although there are periods where (longer) bursts of electrical activity coincide with pyroclasts being ejected to greater heights, at other times the opposite seems true.

## Discussion
Ice nucleation can enhance the amount of charge in the plume, resulting in more and stronger lightning at high altitudes during evolved phases of the eruption, also known as plume volcanic lightning. Recent examples of major eruptions where ice-rich plumes generated a great amount of lightning include the 2018 Anak Krakatau eruption in Indonesia[16], the 2020 Taal eruption in the Philippines[28], and the 2022 Hunga eruption in Tonga[29,30]. Volcanic ash emission during the 2021 Tajogaite eruption was observed to be of variable intensity but the eruption plume height never exceeded the -10 °C isotherm during the time of monitoring, with exception of the stronger explosive event on the evening of 13 December (Fig. 2a). As volcanic ash becomes an effective catalyst for ice nucleation at temperatures below −20 °C[18], it can be concluded that ice nucleation did not play a key role as a plume electrification mechanism during the whole eruption. Similar findings were reported for hundreds of relatively small-scale ash-rich explosive events (<6 km plume height) at Sakurajima volcano, Japan[23]. Nonetheless, as near-vent volcanic lightning was frequently detected at both volcanoes, the importance of silicate particle charging as the responsible plume electrification process is beyond doubt.

The final explosive activity on 13 December 2021 lasted for a few hours and produced a plume that reached a height of ~7600 m a.s.l., exceeding the −20 °C isotherm. At this stage, ice nucleation on ash particles was possible, potentially enhancing plume electrification. However, the electrical discharge rate remained relatively low compared to previous explosive phases, even though the discharge rate did increase in the minutes following the onset of this more vigorous activity (Fig. 2c). The BTD recorded many

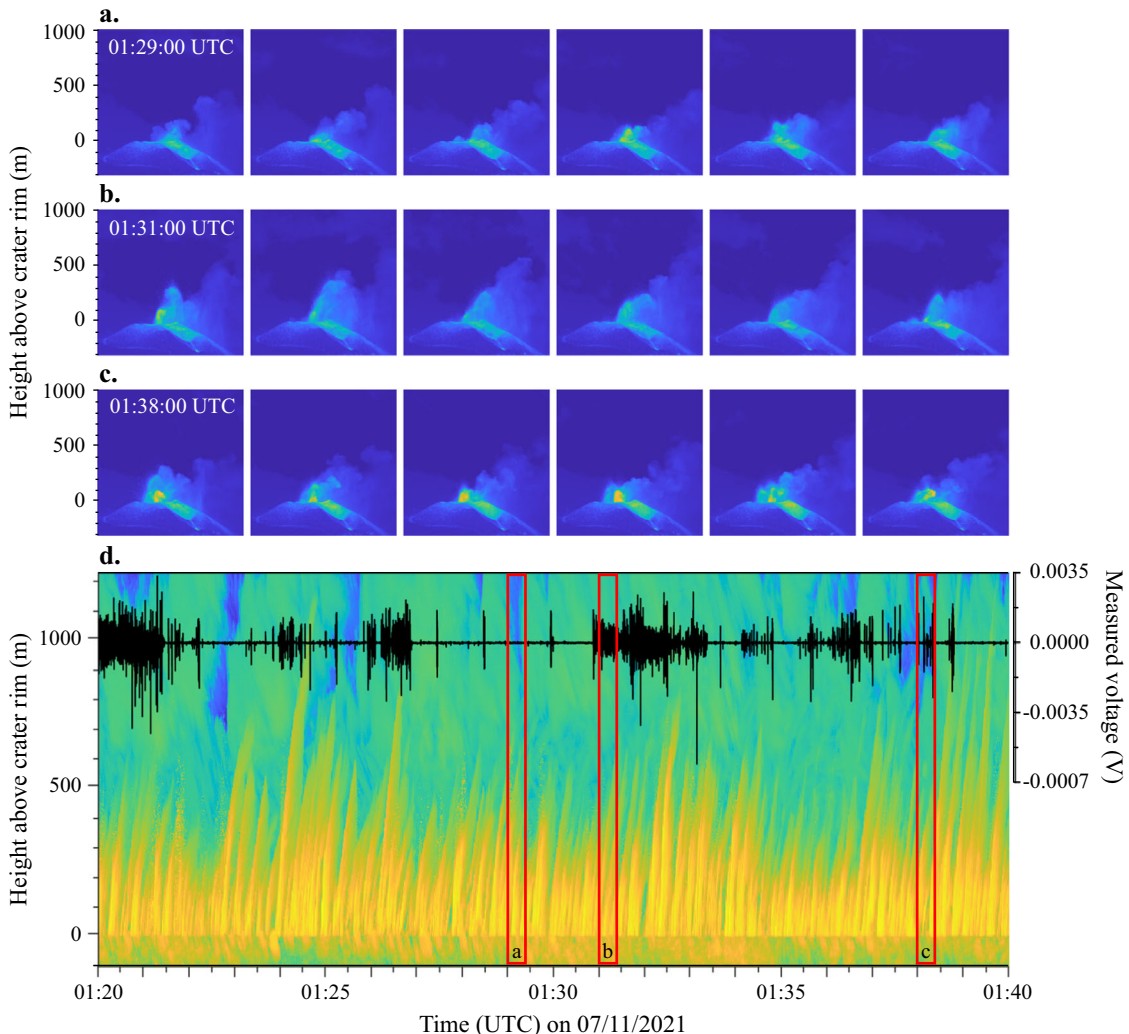

**Fig. 6 | Strombolian activity producing incandescent bombs and a small semi-opaque ash plume on 7 November.** Single frames extracted every 5 seconds from a thermal video taken at 15 fps, with the first frame taken at **a** 01:29:00 UTC, **b** 01:31:00 UTC and **c** 01:38:00 UTC on 7 November 2021. **d** Thermal infrared rise diagram showing the evolution of the maximum temperature (blue is cold and yellow is hot) as a function of time and height (left y-axis). The crater rim of the main active vent is at a height of 0 m in the diagram. In addition, the voltage (V) measured by the primary antenna of BTD2 (black line, right y-axis) is shown, predominantly displaying a mixture of signal types 2 and 3. The red rectangles indicate the time periods of each sequence of thermal frames shown in panels **a-c**.

individual high-amplitude electrical discharges (signal type 1) during this time, but some of these transients had a negative covariance. Particularly the strongest electrical discharges were recorded with an offset of 10–20 ms between the primary and secondary antenna, while for the smaller electrical discharges the secondary antenna did not record any change in the electrostatic field. Based on the electrical signal of type 6 that was recorded earlier that afternoon, we speculate that this is caused by a change or lag in the capacitance of one of the antennas due to ash having been deposited on the sensor. For example, ash covering the shielding cap of the secondary antenna would both increase its capacitance as well as its shielding ability, making the secondary antenna less responsive to fast electrostatic field changes. This false negative covariance affects the detection algorithm which artificially underestimates the electrical discharge rate. Hence, it is possible that ice nucleation aided the generation of volcanic lightning at the end of the eruption, but near-vent charging of silicate particles remains the dominant charging mechanism during the whole Tajogaite eruption, regardless of plume height and eruption style.

Behnke et al.[31] proposed that fluctuations in the electrical discharge rate throughout a single eruption can be either caused by intensified charging of particles at the vent due to an increase in source flux or enhanced plume electrification due to ice nucleation. However, we also observed variations in the electrical activity, and thus the electrical discharge rate, with changing explosive eruption style. This is corroborated by the fact that the highest discharge rates (>5000 discharges per hour) were detected during various eruption styles and there is no correlation between the measured voltage and the electrical discharge rate, as is evident from the normalised discharge rate (Fig. 2b, c). This demonstrates that during the Tajogaite eruption, the fluctuations in the electrical discharge rate are controlled by changes in the mass eruption rate[13,22] as well as changes in eruption style. The former fits with the observation that the fluctuations in the electrical discharge rate are sometimes for short periods of time positively correlated with changes in the plume height (e.g. 29 September–4 November), while the latter explains why high electrical discharge rates can be detected even during times of relatively low volcanic plume height (e.g. 1–3 December). In general, however, no clear correlation was found between the plume height and the electrical discharge rate (Figs. 2a, c), which may have several reasons. It is important to take into account the difference in temporal resolution between the continuous electrical measurements and the plume heights that are reported only a few times per day. Moreover, the Toulouse VAAC focuses on the flight levels that are affected by ash, which may have resulted in an overestimation of the eruption column height, as detached ash clouds can remain at high altitudes for long periods of time. On the other hand, the

plume heights have not been corrected for wind. Field campaign observations showed that the wind could strongly bend the plume, which would create an underestimate of the eruption magnitude[6]. In addition, it is possible for new explosions pulses to eject pyroclasts into an already existing plume without affecting the height of the eruption column, while enhancing the amount of plume electrification and possibly increasing the electrical discharge rate. Moreover, fissure-fed eruptions, such as the Tajogaite eruption, are characterized by different explosive eruption styles occurring simultaneously from multiple vents. This complicates the deconvolution and interpretation of electrical and volcanic tremor signals generated at different vents or in different eruption columns. This can further explain periods where there is little correlation between electrical discharge rate, volcanic tremor amplitude, and plume height (Fig. 2).

The Tajogaite eruption was characterised by a variety of explosive eruption styles: mild and strong ash emission, gas jetting, Strombolian explosions, and lava fountaining. Variations in the electrical signals recorded by the BTD can be linked to distinct changes in the explosive activity. However, to understand the electrical signature of each eruption style, several eruption parameters need to be considered, including the grain size distribution and kinetic energy of the ejected tephra, mass eruption rate, plume height, the temperature of the erupted mass, and the lifetime of the eruption column.

Phases of strong ash emission and lava fountaining, producing ash plumes of several kilometres in height, generated individual high-amplitude electrical discharges (>0.01 V, signal type 1; Fig. 4a-b). Although there are differences between these two explosive eruption styles, both ejected a large amount of ash for hours without interruption, which is responsible for generating substantial charge in the plume through the fracturing and collisions of particles. The development of a tall volcanic ash plume, especially one that is sustained for several hours, allows the build-up of a strong electric field through charge separation and the formation of charge clusters[13,32,33], generating volcanic lightning of high measured voltage. Other explosive activity during the Tajogaite eruption, such as gas jetting, Strombolian activity, or lava fountaining ejecting significantly less ash, lacked the formation of such a well-developed ash plume. This shows that the ejection of large quantities of ash, subsequently undergoing convection and rising to form a mature plume, forms the foundation for the generation of type 1 electrical signals. Similar electrical activity was detected during the impulsive Vulcanian explosions at Sakurajima volcano, Japan[23], and larger-scale explosions at Stromboli volcano, Italy[24]. This suggests that the electrical signals detected during type 1 activity are produced by the commonly observed volcanic lightning, known as near-vent and plume volcanic lightning[34,35]. This is confirmed by direct observations made on the evening of 3 November (Figs. 4a-1). The movement of strongly electrified ash plumes resulted in slow-varying (~1-10 seconds) electrical signals at the antennas (Fig. 3e).

A special case is the sudden onset of strong ash emission in the afternoon on 4 November, which generated high-amplitude electrical discharges almost instantly and was accompanied by abrupt changes in the tremor amplitudes and the values of PC1 and PC2 (Fig. 4). We speculate that this swift change in volcanic activity observed on 4 November was the result of a partial collapse of the shallow plumbing system, which would cause recycled ash to be ejected in addition to juvenile material. Due to the increased activity at the vent, the source of volcanic tremor would have become shallower. In addition, the widening of the conduit would have made the tremor generation mechanism less efficient, leading to a decrease in the volcanic tremor amplitude, consistent with our measurements. Bonadonna et al. (2022)[6] demonstrated the relationship between the stratigraphy, which is directly related to the eruption style, and the variations in volcanic tremor both in terms of amplitude and ratio, which are reflected in PC1 and PC2, respectively. Between 1-3 November, a clear shift in the stratigraphy (from Middle Unit MU2-5 to MU6) was accompanied by a similar, slightly larger decrease in the VLP amplitude and an increase in LP/VLP ratio[6] in comparison to the event on 4 November. Moreover, Middle Unit MU6 contains oxidised red, dull-looking scoriae clasts besides juvenile clasts[6], which could

indeed be an indication of recycled material. Although few ash particles, and thus charge, lingered in the air before the onset of this ash-rich phase, volcanic lightning was generated almost instantly. The charging of recycled ash is mostly limited to particles colliding with each other, as it does not undergo fragmentation like juvenile material. Hence, if our speculation is correct, the occurrence of volcanic lightning during this event shows that instability of the upper volcanic conduit and the crater walls can generate adequate charging by remobilisation and recycling of older incoherent material in addition to that provoked by the ejection of juvenile tephra.

In contrast, mild ash emission producing small ash plumes with a basal diameter of <100 metres (Figs. 4a-2) generated only weak electrical discharges (<0.002 V, signal type 4; Fig. 3d), which predominantly remained undiscovered by the detection algorithm due to the low signal-to-noise ratio of both antennas. This type of explosive activity is driven by much lower kinetic energy and mass eruption rates. Therefore, little plume electrification occurs and consequently few electrical discharges are generated. Signal types 1 and 4 can be viewed as opposite end-members of a continuous spectrum covering a large range of amplitudes and frequencies for individual electrical discharges, depending on the source parameters of the ash-rich explosive events. During the few occasions that movement of charge was detected, the slow-varying electrical signals had a much lower magnitude compared to those detected during larger-scale ash plumes produced by Volcán de Tajogaite, further demonstrating the presence of a weakly charged plume.

Bursts of quasi-continuous electrical discharges of low to intermediate measured voltages (<0.01 V, signal types 2 and 3) were detected during phases of gas jetting (Figs. 4a-3), Strombolian activity (Fig. 6) and lava fountaining (Figs. 4a-5), during which the emission of ash was reduced and the development of a large volcanic ash plume remained absent. This type of electrical activity has not been detected before by BTD measurements at other volcanoes and is distinctly different from the individual high-amplitude electrical discharges detected during strong ash emissions[23,24]. The quasi-continuous electrical discharges are likely the result of different fragmentation efficiencies, eruption dynamics, and source parameters characterising these explosions. Transitions between gas jetting, Strombolian explosions, and lava fountaining are a result of variable conditions of two-phase flow coupling between the magma and the gas in the shallow conduit. These variations in association with relative variation in the magma viscosity (mainly as a function of crystallization rate at shallow depth) regulate the efficiency of the magmatic fragmentation[8,36], therefore affecting the total grain size distribution and total mass of the ejected tephra. It was suggested that secondary brittle fragmentation can occur during lava fountaining when rapid adiabatic cooling is paired with continued gas exsolution and high vesicularity[37], resulting in additional production of fine pyroclasts. Polymodal grain size distribution in the volcanic jet is important to produce charge in the eruption column, as solid particles are the main carriers of charge, and inertia of clasts with different sizes will enhance particle collision and clustering (of both particles and charge) in the turbulent flow[13,38]. During the field campaign in November 2021, production of ash particles was observed during phases of gas jetting, Strombolian activity and lava fountaining (Figs. 4a and 6b), underpinning the possibility of charge generation. The thermals associated with these explosive styles were of short duration so that the eruptive column was mainly limited to the gas-thrust phase, resulting in eruption column heights of only a few hundred metres up to 1 km a.c.r. depending on the eruption style. Moreover, the eruption columns were short-lived due to a large proportion of ejected particles falling back down close to the vent[39], although overlap of falling and rising pyroclasts ejected during the next pulse occurred regularly[8]. Consequently, these types of explosive eruption styles did not build up strongly electrified, convecting plumes with large charge clusters, and as a result, did not generate the conventional volcanic lightning as in the case of sustained ash-rich eruptive episodes. Taddeucci et al.[8] found that although all explosive activity was pulsating, gas jetting (named ash-poor jets in the study), Strombolian activity (spattering), and lava fountains had shorter pulse intervals than the strong ash emissions (ash-rich jets). Also the

maximum particle ejection velocity (MPEV) was higher for these three explosion styles in comparison to strong ash emissions[8]. So rather, the short-lived eruption columns but quickly pulsating nature and high particle ejection velocity of these explosions likely resulted in faster charging and more efficient discharging due to increased turbulence, explaining these bursts of low-amplitude electrical discharges. Interestingly though, the case study shows that lava fountains can generate electrical signals of type 1 as well as types 2 and 3, suggesting that this activity style covers a wide range of eruption dynamics and source parameters. The key difference that determines what type of electrical signal is generated during lava fountaining (type 1 versus types 2 and 3), is the amount of volcanic ash that is being ejected and whether this results in a well-developed plume or not (Supplementary Fig. 5). The difference between signal types 2 and 3 seems more complex, as they share similar characteristics and can occur interchangeably on short time scales, but quantifying this will require detailed time series of the source parameters during future eruptions.

The duration of the electrical bursts increased from gas jetting (<45 seconds; Supplementary Fig. 4e-f) to pulsating Strombolian activity (seconds to minutes) to continuous lava fountaining (predominantly minutes; Supplementary Fig. 4i), which is positively correlated to a combination of relative kinetic energy, mass flux, height of the eruption column and duration of each individual explosion phase. The lava fountains were driven by the highest kinetic energy and were estimated to have the greatest mass flux, ranging between $0.8–2.8 \times 10^4$ kg s$^{-1}$[18], which was reflected in their relatively high eruption columns[40,41]. A high mass flux and MPEV (~24–100 m s$^{-1}$) provide favourable conditions to rapidly generate a lot of plume electrification through a large number of particles fracturing and colliding at a fast rate, which could explain the overall long duration of these bursts. Short term variations most likely affected the efficiency of charging and discharging[31], which could be the reason for shorter bursts interrupting the predominantly minutes-long bursts of electrical activity. In contrast, Strombolian activity was found to have both a relatively intermediate mass flux of $4–9 \times 10^3$ kg s$^{-1}$ and an intermediate MPEV of ~26–32 m s$^{-1}$[18]. The same study also showed that this type of activity consisted for ~18–31% of bombs with a diameter >0.5 m. The presence of fine particles is key for the generation of charge[13,14]. Hence, fluctuations in the grain size distribution, such as the proportion of large bombs, could have either hindered or promoted the plume electrification processes and thus affecting the duration of the electrical bursts. In addition, small variations in the size of the eruption column, as observed in the thermal data (Fig. 6), may further affect the duration, although this relationship is not always evident and requires further investigation. The mass flux of gas jetting was estimated to be the lowest of these three eruption styles, ranging between $0.2–8 \times 10^3$ kg s$^{-1}$[19], creating a relatively low-density eruption column as particles would reach similar heights as in lava fountains. A lower density would mean fewer particles colliding and therefore less charge creation, which could explain why gas jetting generated the shortest bursts of electrical activity. Moreover, the thermal images showed that the core temperature during phases of Strombolian activity and lava fountaining was often higher in comparison to the material ejected during gas jetting. Stern et al.[42] carried out rapid decompression experiments at temperatures up to 320 °C. Although the effect of temperature in the rapid decompression experiments is of difficult interpretation, the results showed that experiments at higher temperatures promoted the increase in the number of small electrical discharges in the gas-particle mixture as well as the total duration of electrical discharges. Furthermore, the experiments at higher temperatures produced the highest charging rates early on in the experiment, this effect being correlated with the increased expansivity and turbulence in the particle-laden jets. This could additionally explain the increased electrical activity during Strombolian activity and lava fountaining. Similarly, a short period of an increased amount of hot material being ejected up to greater heights during gas jetting resulted in longer bursts (comparing Supplementary Fig. 4e–f). This suggests that the temperature of the gas-particle mixture may play an important role in generating the conditions for the occurrence of these bursts of electrical activity as well. All these findings together suggest that signal types 2 and 3 are part of a continuous spectrum, where the duration and magnitude of these bursts depend on many different factors.

The 2021 Tajogaite eruption provided a unique electrical data set, which enabled us to link variations in the electrical activity to changes in the explosive activity during a prolonged eruption. These findings could aid other geophysical parameters in the classification of the different explosive eruption styles, which was particularly challenging for this eruption due to the rapid transitions from one activity into another[8]. A deeper understanding of these electrical signatures and the underlying charge mechanisms could possibly provide estimates of the relative proportion of ash, the mass eruption rate and plume height in the future. This will require further investigation during upcoming eruptions, where detailed time series of the source parameters are obtained and correlated to the electrical signatures. Moreover, our results show that local electrical monitoring of active volcanoes can provide valuable near real-time information on changes in the explosive eruption style as well as the magnitude, which may pass unobserved by regional and global lightning detection systems. The occurrence of volcanic lightning is an indicator of explosive volcanic activity without requiring the need for visibility on the crater. Hence, including electrical detectors in local monitoring networks will become increasingly more important as our ability to interpret these signals improves.

## Methods
### Electrical measurements
The electrical activity generated by the explosive activity of the Tajogaite eruption was recorded by a Biral Thunderstorm Detector BTD-200 (BTD). The sensor was installed on 11 October 2021 at 2.65 km distance NNW from the active craters (location BTD1) and moved SW from the eruptive vents at a distance of 1.77 km (location BTD2) on 27 October 2021 to have a higher detection efficiency as a result of being closer, as well as for logistical reasons. It recorded at location BTD2 until the end of the eruption (Fig. 1). Both installations were located within the exclusion zone of the eruption, which helped reduce the anthropogenic background noise near the instrument.

The BTD measures the slow temporal variation in the electrostatic field within a frequency range of 1–45 Hz[43]. It consists of a primary antenna, which has the highest sensitivity, and a secondary antenna that is shielded by a plastic cap. These antennas allow for the detection of electrical discharges, movements of charge, and impact transients, such as charged precipitation or ash falling on the sensor[23,24,43]. On the one hand, lightning produces transients of the same polarity at both antennas, resulting in a positive covariance between the two signals. On the other hand, charged particles impacting the primary antenna induce a signal of opposite polarity at the secondary antenna, thus resulting in a negative covariance. The raw voltage output from the BTD was digitised using an analog-to-digital converter into a voltage used for calculation by the internal processors[25]. Note that the resulting measured voltage is proportional to the rate of change in the electrostatic field experienced by the antennas, not the voltage of the discharge source itself. The antennas have a saturation level that corresponds to a measured voltage of 0.785 V.

A volcanic lightning detection algorithm, described in Vossen et al.[23], used several empirical thresholds to identify signals as electrical discharges. First, the electrical signals needed to have the same polarity at both antennas and a positive covariance of ≥1.0. Additionally, the ratio between the two antenna signals needed to be >3.0, while the signal-to-noise ratio of the primary and secondary antenna signals needed to be above 2.3 and 1.5, respectively. The covariance and background noise values were calculated over a moving window of 16 and 128 samples, respectively, with a step size of 1 sample.

From the number of electrical discharges identified by the algorithm, the electrical discharge rate (discharges per hour) was calculated. In addition, the average and maximum value of the absolute voltage (V) measured by the primary antenna of the BTD were calculated per hour. To show the relationship between the electrical discharge rate and the amplitude of the

discharges, we normalised the electrical discharge rate by normalising the maximum measured voltage by the saturation level of the primary antenna (0.785 V) and multiplying this with the electrical discharge rate. A 1:1 ratio between the normalised and calculated electrical discharge rate indicates that the strongest electrical discharge within that hour saturated the primary antenna, while a small ratio indicates that the strongest discharge was relatively low in amplitude. Although the electrical parameters are calculated and provided for both BTD locations, these cannot be compared to each other. At frequencies <100 Hz, the electric field decreases proportional to the distance cubed[26,27]. This distance depends on the distance between the BTD and the active vents, the height of the electrical discharges within the plume, and the height and movement of the charged plume with respect to the sensor. BTD2 had a higher detection efficiency due to it being installed 880 m closer to the active vents. As a result, the electrical signals detected by BTD2 typically have a higher amplitude, and thus generally also a higher signal-to-noise ratio, which facilitated the identification of electrical discharges by the algorithm. For this reason, this study focuses mainly on the measurements of BTD2.

The electrical measurements were compared to the varying plume heights and explosive eruption styles using thermal and visual imaging. Note that besides a single flash at 85 km distance from the active craters on 22 November and a thunderstorm on the night of 25–26 November, there is no sign of electrical activity associated with this eruption in the WWLLN dataset. Also, the other global lightning networks did not report any volcanic lightning throughout the event.

### Thermal imaging

To gain more insight into the frequently changing explosive activity of Cumbre Vieja, the continuously recorded electrical data was complemented with thermal videography through the temporary installation of an InfraTec HD thermal infrared (TIR) video camera. The TIR camera was installed NNW from the active craters at a distance of 4.3 km (Fig. 1). The camera was focused on the explosive activity at the eruptive vents. It was recorded almost continuously during a field campaign from 3–8 November with a maximum definition of 640 × 480 pixels at 15 frames per second (fps). A Jenoptik IR 1.0/30 LW objective was used, resulting in a pixel resolution of ~3.6 m at the active vents. The camera software corrected the effects of atmospheric absorption in situ, based on temperature, air humidity, and distance between the camera and the active craters.

We use single frames of the thermal recording to determine the eruption style and time/height thermal infrared diagrams (rise diagrams) to distinguish individual ejection pulses both night and day[25]. TIR rise diagrams show the evolution of the maximum temperature anomaly as a function of time and height[44,45]. To obtain these diagrams, the algorithm developed by Gaudin et al.[44,45] was used, which retains the maximum temperature of each row of a single frame after removing the background brightness by subtracting the previous frame. This analysis was carried out for every 30th thermal frame for a 5-minute time window and every 120th frame for a 20-minute time window (i.e. every 2 and 8 seconds of recording, respectively). In this study, the rise diagrams are used to investigate the link between the electrical signals and the pulsating explosive activity and changes thereof. More detailed future analysis could provide information on the erupted products (ash- or bomb-dominated) and the rise velocity (based on the slope of the traces) as well[44,45].

### Seismic tremor measurements

Seismic tremor measurements were obtained every 50 seconds using seismic station PLPI (Fig. 1a), which was operated by Instituto Volcanológico de Canarias (INVOLCAN), to gain more insight into the processes occurring inside the conduit and at the vents[46]. In this work, we consider the volcanic tremor amplitude in the Very Long Period (VLP, 0.4–0.6 Hz) and the Long Period (LP, 1–5 Hz) frequency bands[25]. Due to the different wavelengths of these components, they provide information about the tremor source mechanism at different depths. Using the local S-wave velocity model of D'Auria et al.[46], we can state that the penetration depth of the VLP component inside the conduit is of few hundred metres, while that of the LP components is a few tens of metres. Therefore, the LP component is more tightly related to the explosive mechanism at the vent, while the VLP component reflects the overall amount of gas flowing through the conduit. Bonadonna et al.[6] demonstrated that the absolute tremor amplitude is related to the intensity of the explosive activity and the ratio between the components reflects changes in the source mechanism of the volcanic tremor and, similarly, in the eruptive mechanism. This can be explained considering that the volcanic tremor wavefield is composed dominantly of Rayleigh waves. Since the volcanic tremor amplitude is highly variable, instead of using the raw ratio, we analysed the temporal variation of these two different components using the Principal Component Analysis (PCA). Before applying PCA, we normalised the amplitudes by taking the logarithm, subtracting the average, and dividing them by the standard deviation. The result of the PCA is provided in Supplementary Fig. 1 in the Supplementary Methods, which shows that the temporal variation is represented by two components. The first principal component (PC1) is mostly related to the absolute amplitude of the volcanic tremor and thus reflects the intensity of the explosive activity. The second principal component (PC2) mostly depends on the ratio between the LP and VLP amplitude, indicating that it reflects the changes in the volcanic tremor source mechanism, which is in turn connected to the type of volcanic activity.

### Background atmospheric conditions

To determine whether ice charging played a role as a plume electrification mechanism in addition to near-vent silicate particle charging, plume heights were compared to the elevation of the 0 °C, −10 °C, and −20 °C isotherms[47]. The Toulouse Volcanic Ash Advisory Center (VAAC) reported the flight levels affected by volcanic ash based on both satellite data and data from the Volcano Observatory Notice for Aviation. The latter was compiled by the Instituto Geográfico Nacional using a camera of the Instituto Astrofísico de Canarias located 16.5 km north of the active vents at an altitude of 2365 m a.s.l.[48]. These flight levels were converted to plume heights, providing a general trend throughout the course of the eruption[25]. Note, however, that these values provide an upper limit, as detached ash clouds may remain at high altitudes for a long period of time even after the explosive activity at the vents has waned or stopped. Moreover, there might be a delay between the plume height and the time it is reported by the Toulouse VAAC, as information was predominantly provided at regular times during the day (03:00, 09:00, 15:00 and 21:00 UTC). We additionally included plume heights that were reported by Plan de Emergencias Volcánicas de Canarias (PEVOLCA) to provide more detail at times when the Toulouse VAAC did not report any change. These plume heights were obtained using the camera of Instituto Astrofísico de Canarias as well, but only once per day (typically mornings, but plotted here at 12:00 p.m. as a fixed time of the day)[25].

Thermodynamic parameters, such as temperature, pressure, relative humidity as well as wind speed and direction, were obtained from weather balloon profiles twice a day (at 00:00 and 12:00 UTC)[25], which were provided by the University of Wyoming, Department of Atmospheric Science (http://weather.uwyo.edu/). However, these weather balloons were released about 150 km east of Volcán de Tajogaite at Güímar (station nr. 60018) on Tenerife island. To ascertain that the temperature measurements are representative of the conditions on La Palma as well, the data was compared to temperature measurements from two ground weather stations of the State Meteorological Agency (AEMET) of Spain on La Palma: El Paso (844 m a.s.l.) and Roque de los Muchachos (2223 m a.s.l.) (Fig. 1)[25]. Although the temperature variation between night and day is greater for the AEMET ground stations, the overall trend is very similar to the weather balloon data set (Supplementary Figs. 2 and 3 in Supplementary Methods). Therefore, we concluded that the Güímar data is sufficiently accurate to construct the different isotherms over La Palma and are representative of the atmospheric conditions encountered by the Tajogaite ash plumes at higher altitudes.

Stable fair-weather conditions over La Palma recorded by meteorological stations during the whole eruption (with the exception of a single thunderstorm episode on 25 and 26 November 2021), allow a confident

**Article**

attribution of changes in lightning activity to the variable explosive activity of Volcán de Tajogaite. WWLLN reported 21 lightning flashes within 20 km radius of La Palma and 886 flashes within 100 km radius between 25 and 26 November. The first and last flashes were detected on 25 November at 16:43:47 UTC and 26 November at 10:48:51 UTC, respectively. During this time, it is unknown whether the electrical discharges detected by the BTD are to be related to volcanic or meteorological lightning.

## Data availability
All data is available here: Vossen, Caron E.J.; Cimarelli, Corrado; D'Auria, Luca; Cigala, Valeria; Kueppers, Ulrich; Barrancos, José; Bennett, Alec J. Multiparametric measurements of the 2021 Tajogaite eruption on La Palma, Canary Islands, Spain. GFZ Data Services. https://doi.org/10.5880/fidgeo.2024.002 (2024).

## Code availability
The custom code used in this study is available upon request from the corresponding author.

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

## Acknowledgements
We thank M. Häberle from Ludwig-Maximilians-Universität (LMU) of Munich for providing technical support and help with the electrical system for our sensor. The authors are thankful to I. Haarer and W. Stoiber for their help in collecting and visualising the standard atmospheric measurements and processing the thermal data. C.C. acknowledges financial support from the Deutsche Forschungsgemeinschaft (German Research Foundation) grant CI 254/2-1 and the ERC Consolidator Grant "VOLTA" under contract N° 864052. V.C. acknowledges financial support from project CI 306/2-1 of the Deutsche Forschungsgemeinschaft. C.V. acknowledges the LMUexcellent PostDoc Support Fund for covering part of the Open Access fee. The authors are grateful to two anonymous reviewers who helped improve the manuscript.

## Author contributions
Caron E.J. Vossen designed the study, provided the methodology and designed the software of the electrical sensor, carried out maintenance, collected, processed and analyzed the electrical and thermal data, visualised the results, and wrote the original draft. Corrado Cimarelli designed the study, installed the electrical sensor, and acquired funding. Luca D'Auria installed the seismic station and processed and analyzed the volcanic tremor measurements. Valeria Cigala recorded and processed the thermal data. Ulrich Kueppers installed the electrical sensor and collected the data. José Barrancos installed the electrical sensor, carried out maintenance, and collected the data. Alec J. Bennett provided the methodology and designed the software of the electrical sensor. All authors revised the manuscript and contributed to the discussion.

## Funding

## Competing interests
The authors declare no competing interests.
