## [Peer Review File · Communications Earth & Environment]

18th Dec 23

Dear Dr Vossen,

Your manuscript titled "Explosive eruption style modulates volcanic electrification signals" has now been seen by 2 reviewers, and we include their comments at the end of this message. They find your work of interest, but some important points are raised. We are interested in the possibility of publishing your study in *Communications Earth & Environment*, but would like to consider your responses to these concerns and assess a revised manuscript before we make a final decision on publication.

We therefore invite you to revise and resubmit your manuscript, along with a point-by-point response that takes into account the points raised. Please highlight all changes in the manuscript text file.

Please use the following link to submit your revised manuscript, point-by-point response to the referees' comments (which should be in a separate document to any cover letter), a tracked-changes version of the manuscript (as a PDF file) and the completed checklist:

[link redacted]

We hope to receive your revised paper within six weeks; please let us know if you aren't able to submit it within this time so that we can discuss how best to proceed. If we don't hear from you, and the revision process takes significantly longer, we may close your file. In this event, we will still be happy to reconsider your paper at a later date, as long as nothing similar has been accepted for publication at *Communications Earth & Environment* or published elsewhere in the meantime.

Please do not hesitate to contact us if you have any questions or would like to discuss these revisions further. We look forward to seeing the revised manuscript and thank you for the opportunity to review your work.

Best regards,

Domenico M. Doronzo, PhD
Editorial Board Member
Communications Earth & Environment
orcid.org/0000-0002-6866-8870

Joe Aslin
Senior Editor
Communications Earth & Environment

EDITORIAL POLICIES AND FORMATTING

Editorial Policy: Policy requirements (Download the link to your computer as a PDF.)

Furthermore, please align your manuscript with our format requirements, which are summarized on the following checklist:

Communications Earth & Environment formatting checklist

and also in our style and formatting guide Communications Earth & Environment formatting guide .

*** DATA: Communications Earth & Environment endorses the principles of the Enabling FAIR data project (<http://www.copdess.org/enabling-fair-data-project/>). We ask authors to make the data that support their conclusions available in permanent, publically accessible data repositories. (Please contact the editor if you are unable to make your data available).

All Communications Earth & Environment manuscripts must include a section titled "Data Availability" at the end of the Methods section or main text (if no Methods). More information on this policy, is available at <http://www.nature.com/authors/policies/data/data-availability-statements-data-citations.pdf>.

If a community resource is unavailable, data can be submitted to generalist repositories such as figshare or Dryad Digital Repository. Please provide a unique identifier for the data (for example a DOI or a permanent URL) in the data availability statement, if possible. If the repository does not provide identifiers, we encourage authors to supply the search terms that will return the data. For data that have been obtained from publically available sources, please provide a URL and the specific data product name in the data availability statement. Data with a DOI should be further cited in the methods reference section.

GENERAL COMMENTS:

Two reviews have now been received, which are detailed and support publication after a similarly detailed work of revision. The reviewers' comments should be addressed and incorporated one-by-one in the revised version. In particular, there are some comments from reviewer 1 (attached pdf), also corroborated by others from reviewer 2, asking to (and I agree with) further using those good data on volcanic lightning detection to be better related with the 2021 Tajogaite eruption style/size or eruption dynamics; this will be useful for volcano monitoring and remote sensing in the future, not only for this volcano but perhaps in general.

Domenico M. Doronzo, PhD
Editorial Board Member
Communications Earth & Environment

REVIEWER COMMENTS:

Reviewer #1 (Remarks to the Author):

This manuscript describes unique measurements of the electrical signals generated by the explosive activity of the 2021 Tajogaite eruption of Cumbre Vieja volcano. The authors recorded a broad variety of very interesting signals that are unequivocally related to the volcanic activity. Given the novelty of the measurements, the complexity of the eruption, and the common occurrence and potential threats of such eruptions, the manuscript represent a potential turning point in our capability to diversify the way in which such eruptions can be monitored remotely. The manuscript should be published timely and in a high-impact, broad-audience journal.

This manuscript also represents the robust foundations on which to build further research, and many of the comments below include ideas and suggestions for follow-up papers that I hope to see published soon.

The interpretation of electrical signals is complex, so more in the case of a complex eruption. The authors effectively rank their findings in decreasing order of robustness, and I follow this useful approach in my comments below.

In order zero there is the fact that eruption-related electrical signals are ubiquitous during the whole eruption, showing that Tajogaite-like eruptions produce electrical signals, and that the signals can be used to assess the beginning and end of eruptions. To my knowledge, local-scale electrical signals associated to mafic eruptions have been reported only once and for a quite different eruption.

The first order is the variability of the signals. The main proposed metric of signal variability, i.e., discharge rate, describes the frequency of peaks in the signal, but: is it effective when the peaks are almost continuous? And then: what about changes in signal amplitude over time? This is currently shown only qualitatively, but why not integrate it in the time series (Fig. 2), perhaps in a way similar to what is done for seismic tremor, by, e.g., doing a RMS time series. Signal classification in 'types' is helpful, but leaves open at least two key questions: 1) how much the types represent end-members of a continuous spectrum rather than well-defined entities with quantifiable differences; and 2) What fraction of the months-long recording each type represent (or, in other words, how often each type recur during the whole eruption).

Third order is the link between electrical signals and changes in explosive activity at the scale of the whole eruption. No obvious link appears between tremor and signals, and this should be mentioned explicitly: negative results are as important as positive ones. At the same scale, the direct correlation between signals and plume height that the authors claim should be investigated a bit more, perhaps

as simply as plotting one parameter against the other.

Fourth order is the link between changes in signals and activity. This is where the paper could be improved most (and a follow-up paper could help). A prerequisite for the signals-activity link is having well-defined qualitative or quantitative descriptors for the activity. The approaches used in figure 4 and, even more, in figure 5, are effective. The latter one offers a beautiful quantitative parallel between electrical signal and changes in activity. If thermal infrared videos are available also for the time interval presented in Fig. 4 or other periods, the same comparison should be presented. Thermal videos showing plumes at different temperatures, as mentioned by the authors, should also be presented. Less effective is the approach followed in the discussion on how styles of activity are linked to signal types. Lava fountaining, for instance, appear to be linked to types 1, 2, and 3, while type 2 signals are attributed to mild ash emissions, gas jetting, Strombolian activity and lava fountaining styles. Such promiscuity questions the effectiveness of styles classification. Why then not rely mainly on other observations mentioned in the manuscript, like the abundance of ash in the plume, jet/bomb height, explosion frequency and so on? If necessary, the authors could then link this qualitative information to more quantitative published data (e.g., mass eruption rate and pyroclast exit velocity) and style names by comparing the abundant visual (photo and video) information already published with their own.

Please find other comments and suggestions in the annotated manuscript.

Reviewer #2 (Remarks to the Author):

This paper presents analysis of new observations of electrical discharges during a recent volcanic event in the Canary Islands. Analysis of concurrent datasets (electrical, seismic, thermal imaging, plume height data) has been carried out to explore the use of electrical measurement as a volcano monitoring technique. This paper provides novel illustration of the potential for electrical monitoring as an eruption monitoring technique, and will be of interest to others in the community and will influence decisions over observations to be made at future eruptions.

Specific points:

1. I would have appreciated a signpost to the 'Methods' section at the beginning of the 'Results' section (i.e. something along the lines of 'Details of the measurement set ups are given in the Methods section) as some acronyms were not defined on first use but instead were subsequently explained in the Methods section. For example, line 260 could do with a reference to Vossen et al and a cross reference to the Methods section; LP and VLP mentioned in line 116 and defined in lines 532&534.
2. Can the authors say more about why the Tajogaite eruption passed under the radar of global lightning networks? (line 63)
3. Line 100 - this isn't clear to me, are you saying that volcanic ash plume heights did not exceed atmospheric freezing levels?
4. Line 126 - can you comment on why the plume heights measured by the different VAAC and PEVOLCA datasets are different (they show similar trends but variation in value)

5. Can you give references for BTM measurements at other volcanoes (line 411)

6. Why was the BTM moved on 26th October? (Line 477)

7. Line 459 onwards serves as a conclusion to the paper and I agree with your conclusions. Is there more you could ask for here, e.g. set out what needs to be done next to understand this further? Should BTMs be deployed at potential eruption sites to give more complete datasets?

Explosive eruption style modulates volcanic electrification signals

Caron E. J. Vossen^{1,*}, Corrado Cimarelli¹, Luca D'Auria^{2,3}, Valeria Cigala¹, Ulrich Kueppers¹,

José Barrancos^{2,3}, Alec J. Bennett^{4,5}

¹ Department of Earth and Environmental Sciences, Ludwig-Maximilians-Universität

München, Munich, Germany

² Instituto Volcanológico de Canarias (INVOLCAN), 38320 San Cristóbal de La Laguna,

Tenerife, Canary Islands, Spain

³ Instituto Tecnológico y de Energías Renovables (ITER). Polígono Industrial de Granadilla,

11 s/n 38600 - Granadilla de Abona, Santa Cruz de Tenerife, Spain

⁴ Bristol Industrial and Research Associates Ltd (Biral), Unit 8 Harbour Road Trading Estate,

Portishead, Bristol, BS20 7BL, United Kingdom

⁵ Department of Electronic and Electrical Engineering, University of Bath, Bath, United

Kingdom

* Corresponding author. E-mail address: caron.vossen@min.uni-muenchen.de

**Abstract**

Volcanic lightning detection has proven useful to volcano monitoring by providing information

on eruption onset, source parameters and ash cloud directions. However, little is known about

the influence of changing eruptive styles on the generation of charge and electrical discharges

inside the eruption column. The 2021 Tajogaite eruption (La Palma, Canary Islands) provided

the rare opportunity to monitor variations in the electrical activity continuously over several

26 weeks using an electrostatic lightning detector. We find that throughout the eruption, silicate
particle charging is the main electrification mechanism. Moreover, we show that the type of
electrical activity is closely linked to the explosive eruption style. Fluctuations in the electrical
discharge rates are likely controlled by variations in the mass eruption rate and/or changes in
the eruption style. These findings hold promise for obtaining near real-time information on the
dynamic evolution of explosive volcanic activity through electrostatic monitoring in the future.

**Introduction**

On 19 September 2021 at 14:10 UTC, an eruption started from a fissure on the Western flank
of the Cumbre Vieja volcanic ridge on La Palma (Canary Islands). This eruption took place 50
36 years after the last eruption in 1971^[1]. It had been preceded by signs of inflation and an increase
in seismic activity in both frequency and intensity, which had gradually become shallower
towards the onset of the eruption^[2]. For almost three months, several vents along a NW-SE
aligned fissure (Figure 1) erupted lava and tephra of basanite to tephrite composition^[3,4]. It
resulted in a 12 km² compound 'A'ā and Pāhoehoe lava flow field^[5], which destroyed entire
villages, infrastructure and plantations, and a  187 m tall scoria cone (1071.2 m above sea level
[a.s.l.]) later named "Volcán de Tajogaite"^[6]. The eruption ended on 13 December 2021.
Surveys using Unoccupied Aerial Vehicles (UAVs) provided total volumes of the subaerial
lava flows (including fallout deposits on lava flows) and the submerged lava deltas of
$177.6 \times 10^6 \text{ m}^3$ and $5.1 \times 10^6 \text{ m}^3$, respectively^[5,6,7]. The volume of the scoria cone was estimated
at $36.5 \times 10^6 \text{ m}^3$ ^[6] and the volume of the total tephra blanket was calculated to be approximately
$2 \times 10^7 \text{ m}^3$ ^[8]. Throughout the eruption, the explosive activity varied on the order of hours to
48 days, ranging between mild to strong ash emissions, gas jetting, Strombolian activity and lava
fountaining^[8,9]. On many occasions, volcanic lightning was observed. Note that there is some
debate about how to classify the different eruption styles during this eruption due to the high

variability in activity without clear boundaries between one activity and another^[10]. The
nomenclature used in this study is following Romero et al. (2022)^[9].

Volcanic lightning is frequently observed during ash-rich explosive eruptions. It is interpreted
as a result of electrification and charge separation in the eruption column^[11]. For plumes that
do not reach atmospheric freezing levels, the dominant charging mechanism is silicate particle
charging, through fracturing of^[12,13] and/or collision of particles^[14,15,16,17]. If sufficiently high
plumes are generated, ice nucleation can further enhance the plume electrification during more
evolved stages of the eruption^[18,19]. Generally, volcanic ash becomes an effective catalyst for
ice nucleation between temperatures of -13 °C and -23 °C^[20,21]. The majority of volcanic
lightning studies were focused on major eruptions using data from global lightning networks,
such as Vaisala GLD360 Global Lightning Detection^[19,22,23], Earth Networks Total Lightning
Network^[18] and the Global Volcanic Lightning Monitor of the World Wide Lightning Location
Network (WWLLN)^[22,24]. However, the 2021 Tajogaite eruption passed under the radar of
these networks, demonstrating that local electrical detectors are required^[25,26]. This eruption
provided the opportunity to continuously detect the electrical activity throughout transitioning
eruption styles and intensities, which would vary on the order of hours to days. With the aim
to link different electrical signals to varying explosive activity, electrostatic data was combined
with thermal videography, visual imaging, standard atmospheric measurements and volcanic
tremor measurements.

Here we demonstrate that electrical discharges were generated almost continuously throughout
the time of monitoring and that silicate particle charging was the main driver for plume
electrification during this eruption. In addition, we find that transitions in the explosive activity,
sometimes accompanied by sudden shifts in the seismic tremor amplitude, can be distinguished
based on distinct changes in the electrical signature.

*Figure 1: Map of La Palma with the locations of Cumbre Vieja volcanic ridge and the*
 *instruments. Google Earth satellite images (Imagery date: 7/7/2019 – newer. Data SIO,*
 *NOAA, U.S. Navy, NGA, GEBCO. <http://www.earth.google.com> [24 October 2023]) a) La*
 *Palma, Canary Islands (Spain), showing the locations of the eruptive craters (red circle) and*
 *seismic station PLPI (white star). b) The 2021 Tajogaite lava flow field (red shaded area; data*
 *from the European agency Copernicus Emergency Management Service,*
 *<https://emergency.copernicus.eu/mapping/list-of-components/EMSR546> [24 October 2023])*
 *is shown together with the locations of BTD1, BTD2 and the thermal camera (white stars), the*
 *location of the active vents (red circles) and nearby villages (blue circles).*

**Results**

**Electrical activity, seismic tremor and plume height**

In Figure 2a we compare the height of the plume (Toulouse VAAC and PEVOLCA datasets)
with the time series of atmospheric temperature at the isotherms 0°C, -10°C and -20°C and the
electrical discharge rate (discharges/6 hours) as detected by the BTD and identified by the
volcanic lightning detection algorithm during the observation period. The results show that
volcanic ash rarely exceeded atmospheric freezing levels throughout the eruption. On 28
September, Toulouse VAAC did report the presence of volcanic ash at altitudes above the -
10°C isotherm, but the BTD had not yet been deployed at this time. Between 18:00 and 03:00
UTC on the night of 13-14 December, volcanic ash was observed at its highest level (almost 8
104 km a.s.l.), exceeding the -20°C isotherm, as a result of intense ash-rich lava fountaining. A
105 relatively small increase in the electrical discharge rate was detected in response to this activity.
This explosive phase stopped around 21:30 UTC on 13 December marking the end of the 2021
Tajogaite eruption. Nonetheless, the ash remained suspended at high altitudes for several hours
longer. Toulouse VAAC did not detect any volcanic ash after 15:00 UTC on 15 December.

In general, fluctuations in plume height appear to be positively correlated to changes in the
electrical discharge rate (Figure 2a). At times where Toulouse VAAC did not report any
changes in the plume height, e.g. from 11-17 November, the daily plume height fluctuations
reported by PEVOLCA do show a positive correlation with the changes in the electrical
discharge rate. The highest electrical discharge rate (>5,000 discharges/hour) was detected
during Strombolian activity on 6 November, which was accompanied by a strong increase in
both the VLP and LP amplitude, as is also evident from PC1 (Figure 2).

Both the LP and VLP tremor amplitude varied throughout the eruption (Figure 2b). The sharp
decrease in amplitude on 27 September coincides with a temporary cease in the eruption shortly
after the cone collapse on 25 September^[9,10]. Similarly, the sudden changes in amplitude on 12
and 13 December are related to a short phase of quiescence followed by the highly explosive
phase on the evening of 13 December. The visual comparison of signals in Figure 2 clearly
shows how the first principal component (PC1) is mostly related to changes in the absolute
tremor amplitude (Figure 2c), while the second principal component (PC2) mainly reflects
changes in the volcanic tremor source mechanism, which is in turn connected to the type of
volcanic activity (Figure 2d).

**Figure 2: Electrical, seismic tremor and plume height measurements throughout the 2021**

**Tajogaite eruption. a) Volcanic plume heights as reported by the Toulouse Volcanic Ash**

**Advisory Center (VAAC, black squares) and the Plan de Emergencias Volcánicas de Canarias**

**(PEVOLCA, green circles) are compared to different temperature regions (shades of blue) as**

**a function of height [m] above sea level. The orange vertical lines indicate electrical discharge**

**rates [discharges/6 hours] detected by the BTM and identified by the volcanic lightning**

**detection algorithm between 11 October and 13 December. b-d) Median calculated over six**

134 hours for **b**) LP (dash-dotted line, left y-axis) and VLP (solid line, right y-axis) amplitudes; **c**)
First principal component (PC1); **d**) Second principal component (PC2). The red vertical lines
and shaded area mark a local thunderstorm on 25 and 26 November. The grey shaded areas
denote the periods for which no BTD data is available.

**Types of electrical activity**

A variety of electrical signals was detected throughout the course of the eruption, which would
change frequently on the order of hours. In general, six types of electrical signatures were
observed, hereafter referred to as types 1-6:

- 1. Individual high-amplitude electrical discharges (typically $>0.01\text{V}$, Figure 3a).
- 2. Minutes-long "bursts" of quasi-continuous low-amplitude electrical activity, generally
ranging between $0.001\text{-}0.01\text{V}$ (Figure 3b).
- 3. Seconds-long ($\sim 2\text{-}30\text{ s}$) "bursts" of quasi-continuous low-amplitude electrical activity,
commonly below 0.005V (Figure 3c).
- 4. Faint electrical discharges with a very low amplitude ($<0.002\text{V}$) recorded by the
primary antenna (Figure 3d). The sensitivity of the secondary antenna is too low to
detect any electrical activity. For this reason, these discharges remain mostly undetected
by the volcanic lightning detection algorithm.
- 5. Movement of charge, visible as slow variations ($\sim 1\text{-}10\text{ s}$) in the electrostatic signal.
Simultaneous electrical discharges are superimposed and can still be detected by the
volcanic lightning detection algorithm (Figure 3e).
- 6. Ash fall on top of the sensor can be discriminated from electrical discharges as these
impact transients produce electrical signals of opposite polarities at the two antennas,

resulting in a negative covariance (Figure 3f). Electrical discharges can still be detected
during ash fall, as is shown by the positive covariance at 10:27:56.6 UTC in Figure 3f.

Combinations of different types were frequently observed. In particular, the individual high-
amplitude discharges (type 1 signal) were often accompanied by movements of charge (type 5
signal). Also, mixtures of seconds- and minutes-long bursts of quasi-continuous low-amplitude
electrical activity (type 2 and 3 signals) were detected. Moreover, short periods of ash fall were
regularly detected during the varying types of electrical activity.

*Figure 3: Different electrical signals recorded during the 2021 Tajogaite eruption. Voltage*

*(V) measured by the primary (black line) and secondary (orange line) antennas of the BTD*

*with the corresponding covariance (V^2) between the primary and secondary signals shown*

*below (purple line). Note the different scale on the y-axis for each panel. a) Individual high-*

*amplitude electrical discharges; b) Minutes-long bursts of quasi-continuous low-amplitude*

*electrical activity; c) Seconds-long bursts of quasi-continuous low-amplitude electrical*

activity; **d)** Faint electrical discharges that generally remain unidentified by the detection
algorithm; **e)** Movement of charge, indicated by the slow-varying electrostatic signal. **f)** Ash
falling on top of the sensor, evidenced by the negative covariance. A single electrical discharge
was detected at 10:27:56.6 UTC.

**Linking electrical activity to explosive eruption styles**

During a field campaign in early November 2021, changes in the explosive activity style and
intensity were observed every few hours. Here, we compare in detail the electrical signals and
electrical discharge rate to visual and thermal images as well as the volcanic tremor time series
for the evening of 3 November and the entire day of 4 November (Figure 4). Close-up plots of
the different electrical signatures are provided in Supplementary Figure 3 of the Supplementary
Results.

Between 21:00 UTC on 3 November and 00:03 UTC on 4 November, BTD2 recorded
individual high-amplitude electrical discharges (type 1 signal), which frequently exceeded a
measured voltage of 0.1V (Figure 4b and Supplementary Figure 3a). This electrical activity
was recorded during ash-rich lava fountaining (Figure 4a-1), which produced a 3 km a.s.l. tall
ash plume (Toulouse VAAC data; Figure 2a). At this time, many flashes of volcanic lightning
were detected (Figure 4c).

After 00:03 UTC, the electrical discharges initially decreased in magnitude ($<0.1V$, Figure 4b
and Supplementary Figure 3b) and subsequently also occurred less frequently (Figure 4c). This
gradual decline in the electrical activity corresponds to a decrease in plume height as reported
by Toulouse VAAC (Figure 2a), indicating a waning of the explosive activity. These changes
are accompanied by a minor increase in both tremor amplitudes and the value of PC1, and a
decrease in the value of PC2. Due to the lower magnitude of the electrical discharges, the

movement of charge (type 5 signal) produced by the moving electrified ash plume relative to
the BTM, becomes visible in the electrical data (Supplementary Figure 3b).

Hereafter, the lava fountaining phase stopped and mild ash emission producing a small and
well-defined ash plume (basal diameter <100 m) was observed Figure 4a-2). This was
accompanied by very faint (<0.001V) electrical discharges (Supplementary Figure 3c) that
predominantly remained undetected by the volcanic lightning detection algorithm (type 4
signal). Consequently, the electrical discharge rate is very low during this period (Figure 4c).

The explosive activity subsequently changed into a phase of gas jetting with minor amounts of
particles being ejected (Figure 4a-3). At times, shock waves were visible, indicating gas
expansion velocities above the speed of sound. Between 13:09-14:16 UTC, BTM2 detected ~3-
10 seconds long bursts of quasi-continuous electrical activity with a measured voltage
commonly below 0.001V (type 3 signal; Supplementary Figure 3d). Between 15:10-15:34
UTC, these bursts increased in duration (up to ~30 seconds) and generally had a slightly higher
measured voltage (<0.002V, Supplementary Figure 3e). Simultaneously, an increase in the
VLP amplitude was recorded, resulting in an increase in the value of PC1 and a decrease in the
value of PC2 (Figure 4d-f). During this time, the thermal images show an increment in the
intensity of the gas jetting accompanied by a small increase in the relative temperature of the
erupted material.

*Figure 4: Explosive activity on 3 and 4 November and the corresponding electrical and*
 *seismic tremor signals. a) Pictures of different explosive eruption styles on 3 and 4 November:*
 *(1) Ash-rich lava fountaining at 23:30:53 UTC on 3 November, producing a plume of*
 *approximately 3 km a.s.l. and a flash of volcanic lightning. Courtesy of Francisco Cáceres*

*Acevedo. Reproduced with permission of the copyright holder; (2) Mild ash emissions at 07:44*
*UTC on 4 November, producing a small ash plume; (3) Gas jetting at 13:11 UTC on 4*
*November; (4) Strong ash emissions at 17:08 UTC on 4 November, producing a plume of*
*approximately 3 km a.s.l.; (5) Lava fountaining without the generation of a large ash plume in*
*contrast to (1). Picture was taken at 21:51 UTC on 4 November. Courtesy of Francisco Cáceres*
*Acevedo. Reproduced with permission of the copyright holder. **b-f)** Measurements taken*
*between 21:00 UTC on 3 November until 00:00 UTC on 5 November. Numbered pictures in*
*panel **a** show prevalent activity of the periods 1-5 reported in panels **b-f**. **b)** Voltage (V)*
*measured by the primary (black line) and secondary (orange line) antennas of BTD2. Close-*
*up plots of the different electrical signatures are shown in Supplementary Figure 3. **c)***
*Electrical discharge rate [discharges/5 minutes]. **d)** Seismic tremor amplitude, showing LP*
*amplitude (dash-dotted line) on the left y-axis and VLP amplitude (solid line) on the right y-*
*axis. The peaks were produced by earthquakes. **e)** First principal component (PC1), which is*
*mostly related to changes in the absolute tremor amplitude. **f)** Second principal component*
*(PC2), which mainly reflects changes in the volcanic tremor source mechanism.*

From 15:46 UTC onward, ash fall (along with a few minutes of light rain) was detected at
BTD2 (type 6 signal), based on electrical signals of opposite polarity at the two antennas
(Supplementary Figure 3f) as well as direct observations. During this period, both the tremor
amplitudes and the value of PC1 gradually decreased, while the value of PC2 gradually
increased (Figure 4d-f). Around 16:57:43 UTC, the explosive activity increased in intensity
and changed rapidly from mild to very strong ash emission (Figure 4a-4, 5a-b), as is manifested
in the rise diagram by the increase in the average height and temperature of the eruption column
(Figure 5c). This sudden change in activity was accompanied by a sharp decrease in both tremor
amplitudes and the value of PC1, as well as a distinct increase in the value of PC2 (Figures 4d–

f). Shortly after at 16:57:50 UTC, individual high-amplitude ($>0.1V$) electrical discharges (type
1 signal) were detected (Figure 4b-c and Supplementary Figure 3g), which lasted for about two
249 hours. The ash plume reached a height of approximately 3 km a.s.l. according to Toulouse
VAAC (Figure 2).

Eventually, the explosive activity changed to a lava fountaining phase again (Figure 4a-5),
dominated by the ejection of incandescent bombs. In general, the lava fountains reached heights
of 500-1000 metres above the crater rim, although individual bombs were occasionally ejected
to greater heights. In contrast to the lava fountaining phase on the evening of 3 November,
relatively little ash was emitted. This change in explosive activity coincided with a change in
electrical activity from individual high-amplitude electrical discharges (type 1 signal) to
minutes-long bursts with measured voltages generally between 0.001-0.01V (type 2 signal,
Supplementary Figure 3h). However, contrary to the signal shown in Figure 3b-c, these signals
did not have only positive covariance values, which is one of the criteria of the volcanic
lightning detection algorithm. For this case, the negative covariance could result from the low
sensitivity of the secondary antenna, or it could be caused by an ash-induced change/lag in the
capacitance of one of the antennas due to previous ash fall deposited on the sensor. This does
not affect the detection of the electrical discharges by the BTM but does result in an
underestimate of the electrical discharge rate as the detection algorithm disregards signals with
a negative covariance (Figure 4c). During this lava fountaining phase, the tremor amplitudes
and the values of PC1 and PC2 remained relatively constant with comparison to the previous
explosive phase.

*Figure 5: Thermal infrared imaging of changing explosive eruption style. Single thermal*
 *frame of a) mild ash emissions at 16:55:30 UTC and b) strong ash emissions at 16:59:30 UTC*
 *entering the clouds approximately 900 metres above the crater rim (see Figure 4a-4). The y-*
 *axes correspond to the y-scale of panel c. c) Thermal infrared rise diagram showing the*
 *evolution of the maximum temperature (blue is cold and yellow is hot) as a function of time*
 *and height (left y-axis). The crater rim of the main active vent is at a height of 0 m in the*
 *diagram. At lower heights, currently inactive vents can be seen as thermal anomalies due to*
 *ongoing passive degassing. The increase in magnitude of the explosive activity at 16:57:43*
 *UTC (vertical red dashed line) is closely followed by the onset of individual high-amplitude*
 *electrical discharges (type 1 activity) as recorded by the primary antenna (black line, right y-*
 *axis) of BTD2.*

Strombolian activity was not observed on 3 and 4 November, but was observed for several
 281 hours on the evening of 6 November. Compared to the phases of lava fountaining, the
 282 incandescent bombs were ejected to much lower heights, generally less than 500 metres above
 283 the crater rim. In addition, a partially opaque ash plume was produced (Figure 6a-b). This type
 of activity produced a mixture of seconds-long and minutes-long bursts of electrical activity
 with measured voltages dominantly between 0.001-0.0025V (type 2 and 3 signals, Figure 6c).

*Figure 6: Strombolian activity producing incandescent bombs and a small semi-opaque ash*
 *plume on 6 November. a) Single frame (at 19:18:39 UTC) of thermal video taken at 15 fps. b)*
 *Picture taken at 18:39:44 UTC from the village Las Manchas. The scale bars indicate the size*
 *of the eruption column. c) Voltage (V) measured by the primary (black line) and secondary*
 *(orange line) antennas of BTD2 showing a mixture of type 2 and 3 signals.*

Discussion

Plume electrification mechanism

Ice nucleation can enhance the amount of charge in the plume, resulting in more and stronger
 volcanic lightning at high altitudes during evolved phases of the eruption, also known as plume
 volcanic lightning. Recent examples of major eruptions where ice-rich plumes generated a

great amount of volcanic lightning include the 2018 Anak Krakatau eruption in Indonesia^[18],
the 2020 Taal eruption in the Philippines^[27] and the 2022 Hunga-Tonga Hunga-Ha'apai
eruption in Tonga^[28,29]. Volcanic ash emission during the 2021 Tajogaite eruption was
observed to be of variable intensity but the eruption plume height never exceeded the -10 °C
isotherm, with exception of the stronger explosive event on the evening of 13 December
(Figure 2a). As volcanic ash becomes an effective catalyst for ice nucleation at temperatures
between -13 °C and -23 °C^[20], it can be concluded that ice nucleation did not play a key role
as a plume electrification mechanism during the whole eruption. Vossen et al. (2021)^[25]
reported similar findings for hundreds of relatively small-scale ash-rich explosive events (< 6
309 km plume height) at Sakurajima volcano, Japan. Nonetheless, as near-vent volcanic lightning
was frequently detected at both volcanoes, the importance of silicate particle charging as the
responsible plume electrification process is beyond doubt.

The final explosive activity on 13 December 2021 lasted for a few hours and produced a plume
that reached a height of ~7600 m a.s.l., exceeding the -20 °C isotherm. At this stage, ice
nucleation on ash particles was likely, potentially enhancing plume electrification. However,
the electrical discharge rate remained relatively low compared to previous explosive phases,
even though the discharge rate did increase in the minutes following the onset of this more
vigorous activity (Figure 2a). The BTD recorded many individual high-amplitude electrical
discharges (type 1 signal) during this time, but some of these transients had a negative
covariance. Particularly the strongest electrical discharges were recorded with an offset of 10-
20 ms between the primary and secondary antenna, while for the smaller electrical discharges
the secondary antenna did not record any change in the electrostatic field. Based on the
electrical signal of type 6 that was recorded earlier that afternoon, we speculate that this is
caused by an ash-induced change or lag in the capacitance of one of the antennas due to ash
fall deposited on the sensor. For example, ash covering the shielding cap of the secondary

antenna would both increase its capacitance as well as its shielding ability, making the
secondary antenna less responsive to fast electrostatic field changes. This false negative
covariance affects the detection algorithm which artificially underestimates the electrical
discharge rate. Hence, it is possible that ice nucleation aided the generation of volcanic
lightning at the end of the eruption, but near-vent charging of silicate particles remains the
dominant charging mechanism during the whole 2021 Tajogaite eruption, regardless of plume
height and eruption style.

**Electrical discharge rate versus plume height**

Behnke et al. (2014)^[30] proposed that fluctuations in the electrical discharge rate throughout a
single eruption can be either caused by intensified vent charging due to an increase in source
flux or enhanced plume electrification due to ice nucleation. However, we also observed
variations in the electrical activity, and thus the electrical discharge rate, with changing
explosive eruption style. For example, while the high-amplitude discharges of type 1 are an
indication of intense electrical activity in the plume, these discharges were generated at a
different frequency compared to the low-amplitude bursts of quasi-continuous lightning of type
2 and 3. Hence, during the Tajogaite eruption the fluctuations in the electrical discharge rate
are controlled by changes in the mass eruption rate^[15,24] as well as changes in eruption style.

The former fits with the observation that the fluctuations in the electrical discharge rate often
are positively correlated with changes in the plume height, while the latter explains why high
electrical discharge rates can be detected even during times of relatively low volcanic plume
height. Moreover, fissure-fed eruptions, such as the 2021 Tajogaite eruption, are characterised
by different explosive eruption styles occurring simultaneously from multiple vents. This
complicates the deconvolution and interpretation of electrical and volcanic tremor signals

generated at different vents or in different eruption columns. This can further explain periods
where there is little correlation between electrical discharge rate, volcanic tremor amplitude
and plume height.

The highest electrical discharge rate was measured during an electrical phase of type 2 and 3,
which was produced by Strombolian activity on 6 November (Figure 6). These bursts of quasi-
continuous electrical discharges can result in very high electrical discharge rates. The brief
increase in both VLP and LP amplitudes during this period was probably caused by a rise in
explosive intensity. The higher intensity most likely played an important role in efficient
charging and discharging of the eruption column, resulting in the high recorded electrical
discharge rate.

**Linking electrical signatures to explosive eruption styles**

The Tajogaite eruption was characterised by a variety of explosive eruption styles: mild and
strong ash emission, gas jetting, Strombolian explosions and lava fountaining. Variations in
the electrical signals recorded by the BTD can be linked to distinct changes in the explosive
activity. However, to understand the electrical signature of each eruption style, several eruption
parameters need to be considered, including the grain size distribution and kinetic energy of
ejected tephra, mass eruption rate, plume height, temperature of the erupted mass and the
lifetime of the eruption column.

Phases of strong ash emission or lava fountaining producing ash plumes of several kilometres
in height, generated individual high-amplitude ($>0.01V$) electrical discharges (type 1 signal;
Figure 4a-b). This eruption style is characterised by a high kinetic energy and mass eruption
rate, which is responsible for generating substantial charge in the plume through the fracturing
and collisions of particles. The development of a tall volcanic ash plume, especially one that is

[revised manuscript text omitted]

size distribution in the volcanic jet is important to produce charge in the eruption column, as
solid particles are the main carriers of charge and inertia of clasts with different sizes will
enhance particle collision and clustering (of both particles and charge) in the turbulent
flow^[15,37]. During the field campaign in November 2021, production of ash particles during
phases of gas jetting, Strombolian activity and lava fountaining has been frequently observed
(Figure 4a and 6b). The thermals associated with these explosive styles are of short duration so
that the eruptive column is mainly limited to the gas-thrust phase and does not develop strong
convection as in the case of sustained eruptive episodes. As a result, the eruption columns reach
only a few hundreds of metres in height and are short-lived due to their pulsing behaviour^[38].
Consequently, these types of explosive eruption styles do not build up strongly electrified
plumes with large charge clusters, and as a result do not generate the conventional volcanic
lightning. Rather, the pulsating nature of these explosions likely resulted in faster charging and
more efficient discharging due to increased turbulence, explaining these bursts of low-
amplitude electrical discharges.

The duration of these bursts increased from gas jetting (< 30 seconds) (Supplementary Figure
3d-e) to pulsating Strombolian activity (seconds to minutes) to continuous lava fountaining
(minutes) (Supplementary Figure 3h), which is positively correlated to the relative kinetic
energy, the mass eruption rate, the height of the eruption column and the duration of each
individual explosion phase. The lava fountains are driven by the highest kinetic energy,
reflected in their higher eruption columns^[39,40]. Moreover, the thermal images showed that the
temperature of pyroclasts ejected during phases of the Strombolian activity and lava
fountaining was higher compared to the material ejected during gas jetting. Stern et al.
(2019)^[41] carried out rapid decompression experiments at temperatures up to 320 °C. Although
the effect of temperature in the rapid decompression experiments is of difficult interpretation,
the results showed that experiments at higher temperature promoted the increase in the number

[revised manuscript text omitted]

function of time and height^[43,44]. To obtain these diagrams, the algorithm developed by Gaudin
et al. (2017a,b)^[43,44] was used, which retains the maximum temperature of each row of a single
frame after removing the background brightness by subtracting the previous frame. This
analysis was carried out for every 30th thermal frame (i.e. every 2 seconds of recording). These
rise diagrams provide information on the timing of individual ejection pulses, the erupted
products (ash- or bomb-dominated) and the rise velocity (based on the slope of the traces)^[43,44].

**Seismic tremor measurements**

Seismic tremor measurements were obtained using seismic station PLPI (Figure 1a), which
was operated by Instituto Volcanológico de Canarias (INVOLCAN), to gain more insight into
the processes occurring inside the conduit and at the vents^[45]. Bonadonna et al. (2022)^[8]
considered the volcanic tremor amplitude in the Very Long Period (VLP, 0.4-0.6 Hz) and the
Long Period (LP, 1-5 Hz) frequency bands. They demonstrated that the absolute tremor
amplitude and the ratio between the components were related to the intensity of the explosive
activity and its typology, respectively. This can be explained considering that the volcanic
tremor wavefield is composed dominantly of Rayleigh waves. Using the local S-wave velocity
model of D'Auria et al. (2022)^[45], we can state that the penetration depth of the VLP component
inside the conduit is of few hundred metres, while that of the LP components is a few tens of
metres. Therefore, the LP component is more tightly related to the explosive mechanism at the
vent, while the VLP component reflects the overall amount of gas flowing through the conduit.
In this work, we analysed the temporal variation of these two different components using the
Principal Component Analysis (PCA). Before applying PCA, we normalised the amplitudes by
taking the logarithm, subtracting the average and dividing them by the standard deviation. In

the following, we show how the temporal variation is represented in the two components (PC1
and PC2) retrieved by applying PCA.

**Background atmospheric conditions**

To determine whether ice charging played a role as a plume electrification mechanism in
addition to near-vent silicate particle charging, plume heights were compared to the elevation
of the 0°C, -10°C and -20°C isotherms^[46].

The Toulouse Volcanic Ash Advisory Center (VAAC) reports the flight levels affected by
volcanic ash based on satellite data, together with data from the Volcano Observatory Notice
for Aviation compiled by the Instituto Geográfico Nacional using a camera of the Instituto
Astrofísico de Canarias located 16.5 km north of the active vents at an altitude of 2365 m above
sea level^[47]. These flight levels were converted to plume heights, providing a general trend
throughout the course of the eruption. Note, however, that these values provide an upper limit,
as detached ash clouds may remain at high altitudes for a long period of time even after the
explosive activity at the vents has waned or stopped. Moreover, there might be a delay between
the plume height and the time it is reported by the Toulouse VAAC. We additionally included
plume heights that were reported by Plan de Emergencias Volcánicas de Canarias (PEVOLCA)
once per day (typically mornings) to provide more detail at times where the Toulouse VAAC
did not report any change.

Thermodynamic parameters, such as temperature, pressure, relative humidity as well as wind
speed and direction, were obtained from weather balloon profiles twice a day (at 00:00 and
12:00 UTC), which were provided by the University of Wyoming, Department of Atmospheric
Science (<http://weather.uwyo.edu/>). However, these weather balloons were released about 150
568 km east at Güímar (station nr. 60018) on Tenerife island. To ascertain that the temperature

measurements are representative for the conditions on La Palma as well, the data was compared
to temperature measurements from two ground weather stations of the State Meteorological
Agency (AEMET) of Spain on La Palma: El Paso (844 m a.s.l.) and Roque de los Muchachos
(2223 m a.s.l.) (Figure 1). Although the temperature variation between night and day is greater
for the AEMET ground stations, the overall trend is very similar to the weather balloon data
set (Supplementary Figures 1 and 2 in Supplementary Methods). Therefore, we concluded that
the Güímar data is sufficiently accurate to construct the different isotherms over La Palma and
are representative of the atmospheric conditions encountered by the Tajogaite ash plumes at
higher altitude.

Stable fair-weather conditions over La Palma recorded by meteorological stations during the
whole eruption (with the exception of a single thunderstorm episode on 25 and 26 November
2021), allow a confident attribution of changes in lightning activity to the variable explosive
activity of Volcán de Tajogaite. WWLLN reported 21 lightning flashes within 20 km radius of
La Palma and 886 flashes within 100 km radius between 25-26 November. The first and last
flashes were detected on 25 November at 16:43:47 UTC and 26 November at 10:48:51 UTC,
respectively. During this time, it is unknown whether the electrical discharges detected by the
BTM are to be related to volcanic or meteorological lightning.

**Data availability**

The data that support the findings of this study are available from the corresponding author
upon request.

**Acknowledgements**

We thank M. Häberle from Ludwig-Maximilians-Universität (LMU) of Munich for providing
technical support and help with the electrical system for our sensor. The authors are thankful
to I. Haarer and W. Stoiber for their help in collecting and visualising the standard atmospheric
measurements and processing the thermal data. C.C. acknowledges financial support from the
German Research Foundation grant CI 254/2-1 and the ERC Consolidator Grant “VOLTA”
under contract N° 864052. V.C. acknowledges financial support from project CI 306/2-1 of the
Deutsche Forschungsgemeinschaft (German Science Foundation).

**Author contributions**

Caron E.J. Vossen designed the study, provided the methodology and designed the software of
the electrical sensor, carried out maintenance, collected, processed and analysed the electrical
and thermal data, visualised the results and wrote the original draft. Corrado Cimarelli designed
the study, installed the electrical sensor and acquired funding. Luca D’Auria installed the
seismic station and processed and analysed the volcanic tremor measurements. Valeria Cigala
recorded and processed the thermal data. Ulrich Küppers installed the electrical sensor and
collected the data. José Barrancos installed the electrical sensor, carried out maintenance and
collected the data. Alec J. Bennett provided the methodology and designed the software of the
electrical sensor. All authors revised the manuscript and contributed to the discussion.

**Competing interests**

The authors declare no competing interests.

**References**

1. Longpré, M. A., & Felpeto, A. Historical volcanism in the Canary Islands; part 1: A review
of precursory and eruptive activity, eruption parameter estimates, and implications for
hazard assessment. *J. Volcanol. Geotherm. Res.*, **419**, 107363 (2021).

2. De Luca, C. *et al.* Pre-and Co-Eruptive Analysis of the September 2021 Eruption at
Cumbre Vieja Volcano (La Palma, Canary Islands) Through DInSAR Measurements and
Analytical Modeling. *Geophys. Res. Lett.*, **49**, e2021GL097293.
<https://doi.org/10.1029/2021GL097293> (2022).

3. Castro, J. M., & Feisel, Y. Eruption of ultralow-viscosity basanite magma at Cumbre
Vieja, La Palma, Canary Islands. *Nat. Commun.*, **13**, 1-12. [https://doi.org/10.1038/s41467-](https://doi.org/10.1038/s41467-022-30905-4)
[022-30905-4](https://doi.org/10.1038/s41467-022-30905-4) (2022).

4. Pankhurst, M. J. *et al.* Rapid response petrology for the opening eruptive phase of the 2021
Cumbre Vieja eruption, La Palma, Canary Islands. *Volcanica*, **5**, 1-10.
<https://doi.org/10.30909/vol.05.01.0110> (2022).

5. Román, A. *et al.* Unmanned aerial vehicles (UAVs) as a tool for hazard assessment: The
2021 eruption of Cumbre Vieja volcano, La Palma Island (Spain). *Sci. Total Environ.*, **843**,
157092. <https://doi.org/10.1016/j.scitotenv.2022.157092> (2022).

- 6. Civico, R. *et al.* High-resolution Digital Surface Model of the 2021 eruption deposit of
Cumbre Vieja volcano, La Palma, Spain. *Sci. Data*, **9**, 1-7.
<https://doi.org/10.1038/s41597-022-01551-8> (2022).
- 7. Plank, S. *et al.* Combining thermal, tri-stereo optical and bi-static InSAR satellite imagery
for lava volume estimates: the 2021 Cumbre Vieja eruption, La Palma. *Sci. Rep.*, **13**, 2057.
<https://doi.org/10.1038/s41598-023-29061-6> (2023).
- 8. Bonadonna, C. *et al.* Physical Characterization of Long-Lasting Hybrid Eruptions: The
2021 Tajogaite Eruption of Cumbre Vieja (La Palma, Canary Islands). *J. Geophys. Res.:
Solid Earth*, **127**, e2022JB025302. <https://doi.org/10.1029/2022JB025302> (2022).
- 9. Romero, J. E. *et al.* The initial phase of the 2021 Cumbre Vieja ridge eruption (Canary
Islands): Products and dynamics controlling edifice growth and collapse. *J. Volcanol.
Geotherm. Res.*, 107642. <https://doi.org/10.1016/j.jvolgeores.2022.107642> (2022).
- 10. Taddeucci, J. *et al.* The Explosive Activity of the 2021 Tajogaite Eruption (La Palma,
Canary Islands, Spain). *Geochem. Geophys. Geosystems*, **24**, e2023GC010946.
<https://doi.org/10.1029/2023GC010946> (2023).
- 11. Cimarelli, C. and Genareau, K. A review of volcanic electrification of the atmosphere and
volcanic lightning. *J. Volcanol. Geotherm. Res.*, page 107449.
<https://doi.org/10.1016/j.jvolgeores.2021.107449> (2021).

- 12. Dickinson, J. *et al.* Fractoemission from fused silica and sodium silicate glasses. *J. Vac.*
*Sci. Technol. A: Vac. Surf. Films*, **6**, 1084–1089. <https://doi.org/10.1116/1.575646> (1988).
- 13. James, M., Lane, S., and Gilbert, J. S. Volcanic plume electrification: Experimental
investigation of a fracture-charging mechanism. *J. Geophys. Res.: Solid Earth*, **105**,
16641–16649. <https://doi.org/10.1029/2000JB900068> (2000).
- 14. Lacks, D. J. and Levandovsky, A. Effect of particle size distribution on the polarity of
triboelectric charging in granular insulator systems. *J. Electrostat.*, **65**, 107–112.
<https://doi.org/10.1016/j.elstat.2006.07.010> (2007).
- 15. Cimarelli, C., Alatorre-Ibargüengoitia, M., Kueppers, U., Scheu, B., and Dingwell, D. B.
Experimental generation of volcanic lightning. *Geology*, **42**, 79–82.
<https://doi.org/10.1130/G34802.1> (2014).
- 16. Gaudin, D. and Cimarelli, C. The electrification of volcanic jets and controlling
parameters: A laboratory study. *Earth Planet. Sci. Lett.*, **513**, 69–80.
<https://doi.org/10.1016/j.epsl.2019.02.024> (2019).
- 17. Harper, J. M., Cimarelli, C., Cigala, V., Kueppers, U., and Dufek, J. Charge injection into
the atmosphere by explosive volcanic eruptions through triboelectrification and
fragmentation charging. *Earth Planet. Sci. Lett.*, **574**, 117162 (2021).
- 18. Prata, A. *et al.* Anak Krakatau triggers volcanic freezer in the upper troposphere. *Sci. Rep.*,
**10**, 1–13 (2020).

19. Van Eaton, A. R. *et al.* Did ice-charging generate volcanic lightning during the 2016–2017 eruption of Bogoslof volcano, Alaska? *Bull. Volcanol.*, **82**, 1–23 (2020).
20. Durant, A. J., Shaw, R., Rose, W. I., Mi, Y., and Ernst, G. Ice nucleation and overseeding of ice in volcanic clouds. *J. Geophys. Res.: Atmos.*, **113** (2008).
21. Maters, E. C., Cimarelli, C., Casas, A. S., Dingwell, D. B., & Murray, B. J. Volcanic ash ice-nucleating activity can be enhanced or depressed by ash-gas interaction in the eruption plume. *Earth Planet. Sci. Lett.*, **551**, 116587. <https://doi.org/10.1016/j.epsl.2020.116587> (2020).
22. Haney, M. M. *et al.* Characteristics of thunder and electromagnetic pulses from volcanic lightning at Bogoslof volcano, Alaska. *Bull. Volcanol.*, **82**, 15. <https://doi.org/10.1007/s00445-019-1349-y> (2020).
23. Jarvis, P. A., Caldwell, G., Noble, C., Ogawa, Y., & Vagasky, C. Volcanic lightning reveals umbrella cloud dynamics of the January 2022 Hunga Tonga-Hunga Ha’apai eruption. Preprint at <https://doi.org/10.31223/X5D09J> (2023).
24. Hargie, K. A. *et al.* Globally detected volcanic lightning and umbrella dynamics during the 2014 eruption of Kelud, Indonesia. *J. Volcanol. Geotherm. Res.*, **382**, 81-91. <https://doi.org/10.1016/j.jvolgeores.2018.10.016> (2019).

- 25. Vossen, C. E. J. *et al.* Long-term observation of electrical discharges during persistent
Vulcanian activity. *Earth Planet. Sci. Lett.*, **570**, 117084.
<https://doi.org/10.1016/j.epsl.2021.117084> (2021).
- 26. Vossen, C. E. J. *et al.* The electrical signature of mafic explosive eruptions at Stromboli
volcano, Italy. *Sci. Rep.*, **12**, 1-13. <https://doi.org/10.1038/s41598-022-12906-x> (2022).
- 27. Van Eaton, A. R., Smith, C. M., Pavlonis, M., and Said, R. Eruption dynamics leading to
a volcanic thunderstorm—The January 2020 eruption of Taal volcano, Philippines.
*Geology*, **50**, 491-495. <https://doi.org/10.1130/G49490.1> (2021).
- 28. Ichihara, M., Mininni, P. D., Ravichandran, S., Cimarelli, C., and Vagasky, C. Multiphase
turbulent flow explains lightning rings in volcanic plumes. *Commun. Earth Environ.*, **4**,
417. <https://doi.org/10.1038/s43247-023-01074-z> (2023).
- 29. Van Eaton, A. R. *et al.* Lightning rings and gravity waves: Insights into the giant eruption
plume from Tonga's Hunga Volcano on 15 January 2022. *Geophys. Res. Lett.*, **50**,
e2022GL102341. <https://doi.org/10.1029/2022GL102341> (2023).
- 30. Behnke, S. A., Thomas, R. J., Edens, H. E., Krehbiel, P. R., & Rison, W. The 2010 eruption
of Eyjafjallajökull: Lightning and plume charge structure. *J. Geophys. Res.: Atmos.*, **119**,
833-859. <https://doi.org/10.1002/2013JD020781> (2014).
- 31. Mather, T. A., & Harrison, R. G. Electrification of volcanic plumes. *Surv. Geophys.*, **27**,
387-432. <https://doi.org/10.1007/s10712-006-9007-2> (2006).

32. Cimarelli, C. *et al.* Multiparametric observation of volcanic lightning: Sakurajima Volcano, Japan. *Geophys. Res. Lett.*, **43**, 4221-4228. <https://doi.org/10.1002/2015GL067445> (2016).
33. Thomas, R.J., *et al.* Lightning and Electrical Activity During the 2006 Eruption of Augustine Volcano: Chapter 25 in *The 2006 Eruption of Augustine Volcano, Alaska*. No. 1769-25. US Geological Survey, pp. 579–608. <https://doi.org/10.3133/pp176925> (2010).
34. Cimarelli, C., Behnke, S., Genareau, K., Harper, J. M., & Van Eaton, A. R. Volcanic electrification: recent advances and future perspectives. *Bull. Volcanol.*, **84**, 78. <https://doi.org/10.1007/s00445-022-01591-3> (2022).
35. Cimarelli, C., Di Traglia, F., & Taddeucci, J. Basaltic scoria textures from a zoned conduit as precursors to violent Strombolian activity. *Geology*, **38**, 439-442. <https://doi.org/10.1130/G30720.1> (2010).
36. Namiki, A., Patrick, M. R., Manga, M., & Houghton, B. F. Brittle fragmentation by rapid gas separation in a Hawaiian fountain. *Nat. Geosci.*, **14**, 242-247. <https://doi.org/10.1038/s41561-021-00709-0> (2021).
37. Di Renzo, M., & Urzay, J. Aerodynamic generation of electric fields in turbulence laden with charged inertial particles. *Nat. Commun.*, **9**, 1676. <https://doi.org/10.1038/s41467-018-03958-7> (2018).

- 38. Taddeucci, J., Edmonds, M., Houghton, B., James, M. R., and Vergnolle, S. Hawaiian
and Strombolian eruptions. In *The Encyclopedia of Volcanoes*, pages 485–503. Elsevier
(2015).
- 39. Wilson, L., Sparks, R. S. J., Huang, T. C., & Watkins, N. D. The control of volcanic
column heights by eruption energetics and dynamics. *J. Geophys. Res.: Solid Earth*, **83**,
1829-1836. <https://doi.org/10.1029/JB083iB04p01829> (1978).
- 40. Parfitt, E. A., & Wilson, L. Explosive volcanic eruptions—IX. The transition between
Hawaiian-style lava fountaining and Strombolian explosive activity. *Geophys. J. Int.*, **121**,
226-232. <https://doi.org/10.1111/j.1365-246X.1995.tb03523.x> (1995).
- 41. Stern, S., Cimarelli, C., Gaudin, D., Scheu, B., and Dingwell, D. Electrification of
experimental volcanic jets with varying water content and temperature. *Geophys. Res.*
*Lett.*, **46**, 11136–11145. <https://doi.org/10.1029/2019GL084678> (2019).
- 42. Bennett, A. J. Electrostatic thunderstorm detection. *Weather*, **72**, 51–54 (2017).
- 43. Gaudin, D. *et al.* Integrating puffing and explosions in a general scheme for Strombolian-
style activity. *J. Geophys. Res.: Solid Earth*, **122**, 1860-1875.
<https://doi.org/10.1002/2016JB013707> (2017a).
- 44. Gaudin, D. *et al.* Characteristics of puffing activity revealed by ground-based, thermal
infrared imaging: the example of Stromboli Volcano (Italy). *Bull. Volcanol.*, **79**, 1-15.
<https://doi.org/10.1007/s00445-017-1108-x> (2017b).

45. D'Auria, L., *et al.* Rapid magma ascent beneath La Palma revealed by seismic
tomography. *Sci. Rep.*, **12**, 17654. <https://doi.org/10.1038/s41598-022-21818-9> (2022).

46. Arason, P., Bennett, A. J., and Burgin, L. E. Charge mechanism of volcanic lightning
revealed during the 2010 eruption of Eyjafjallajökull, *J. Geophys. Res.: Solid Earth*, **116**,
B00C03, <https://doi.org/10.1029/2011JB008651> (2011).

47. Felpeto, A., Molina-Arias, A. J., Quirós, F., Pereda, J., & Díaz-Suárez, E. A. Measuring
the height of the eruptive column during the 2021 eruption of Cumbre Vieja (La Palma
Island, Canary Islands). In *EGU General Assembly Conference Abstracts*, pp. EGU22-
9419 (2022).

The reviewers' comments are given in **bold** (with line numbers corresponding to the original draft), author replies in *italics* and manuscript edits in quotation marks and regular text, with corresponding line numbers.

Note that the DOI provided in the Data Availability statement has not been registered yet. All data will be made public upon acceptance of the manuscript. The data is currently available for review with the following temporary link:

<https://dataservices.gfz-potsdam.de/panmetaworks/review/5c0d2f8e3859809de54e1ca112f47558dffc420d571ca339cc6eea543f1f0107/>

Reviewer 1

This manuscript describes unique measurements of the electrical signals generated by the explosive activity of the 2021 Tajogaite eruption of Cumbre Vieja volcano. The authors recorded a broad variety of very interesting signals that are unequivocally related to the volcanic activity. Given the novelty of the measurements, the complexity of the eruption, and the common occurrence and potential threats of such eruptions, the manuscript represents a potential turning point in our capability to diversify the way in which such eruptions can be monitored remotely. The manuscript should be published timely and in a high-impact, broad-audience journal.

This manuscript also represents the robust foundations on which to build further research, and many of the comments below include ideas and suggestions for follow-up papers that I hope to see published soon.

The interpretation of electrical signals is complex, so more in the case of a complex eruption. The authors effectively rank their findings in decreasing order of robustness, and I follow this useful approach in my comments below.

In order zero there is the fact that eruption-related electrical signals are ubiquitous during the whole eruption, showing that Tajogaite-like eruptions produce electrical signals, and that the signals can be used to assess the beginning and end of eruptions. To my knowledge, local-scale electrical signals associated to mafic eruptions have been reported only once and for a quite different eruption.

The first order is the variability of the signals. The main proposed metric of signal variability, i.e., discharge rate, describes the frequency of peaks in the signal, but: is it effective when the peaks are almost continuous? And then: what about changes in signal amplitude over time? This is currently shown only qualitatively, but why not integrate it in the time series (Fig. 2), perhaps in a way similar to what is done for seismic tremor, by, e.g., doing a RMS time series. Signal classification in 'types' is helpful, but leaves open at least two key questions: 1) how much the types represent end-members of a continuous spectrum rather than well-defined entities with quantifiable differences; and 2) What fraction of the months-long recording each type represent (or, in other words, how often each type recur during the whole eruption).

Third order is the link between electrical signals and changes in explosive activity at the scale of the whole eruption. No obvious link appears between tremor and signals, and this should be mentioned explicitly: negative results are

as important as positive ones. At the same scale, the direct correlation between signals and plume height that the authors claim should be investigated a bit more, perhaps as simply as plotting one parameter against the other.

Fourth order is the link between changes in signals and activity. This is where the paper could be improved most (and a follow-up paper could help). A prerequisite for the signals-activity link is having well-defined qualitative or quantitative descriptors for the activity. The approaches used in figure 4 and, even more, in figure 5, are effective. The latter one offers a beautiful quantitative parallel between electrical signal and changes in activity. If thermal infrared videos are available also for the time interval presented in Fig. 4 or other periods, the same comparison should be presented. Thermal videos showing plumes at different temperatures, as mentioned by the authors, should also be presented. Less effective is the approach followed in the discussion on how styles of activity are linked to signal types. Lava fountaining, for instance, appear to be linked to types 1, 2, and 3, while type 2 signals are attributed to mild ash emissions, gas jetting, Strombolian activity and lava fountaining styles. Such promiscuity questions the effectiveness of styles classification. Why then not rely mainly on other observations mentioned in the manuscript, like the abundance of ash in the plume, jet/bomb height, explosion frequency and so on? If necessary, the authors could then link this qualitative information to more quantitative published data (e.g., mass eruption rate and pyroclast exit velocity) and style names by comparing the abundant visual (photo and video) information already published with their own.

Please find other comments and suggestions in the annotated manuscript.

We thank the reviewer for taking the time and effort to go through the manuscript so thoroughly and for providing us with such a positive review and summary of all the comments. It is great to read that the reviewer understands the value of the electrical signals that we present here and how they might help volcano monitoring in the future. We are truly grateful for all the detailed comments and questions, which have significantly improved the manuscript, especially regarding how the results are presented and the way the correlations between the different datasets are discussed. The questions and comments that are included in the review summary have all been answered using the annotations in the manuscript, as provided below.

Line 41: formed

We thank the reviewer for this comment as it shows that the sentence did not read well. We have added the word “formed” and also replaced “it” by “the eruption” at the beginning of the sentence. It now reads:

Lines 37-40: “The eruption resulted in a 12 km² compound ‘A`ā and Pāhoehoe lava flow field^[4], which destroyed entire villages, infrastructure and plantations, and formed a 187 m tall scoria cone (1071.2 m above sea level) later named “Volcán de Tajogaite”^[5].”

Line 67: from a BTD (please define acronyms at first use) stations

The authors thank the reviewer for pointing out that certain acronyms need to be defined earlier. The manuscript was originally written with the Methods before the Results section. We have realised that we forgot to correct the location of a few of

the acronym definitions in the text upon rearranging the order of the sections. As the reviewer suggested, we have now mentioned the name of the electrical sensor together with its acronym in the Introduction section:

Lines 61-64: “With the aim to link different electrical signals to varying explosive activity, electrostatic data from a Biral Thunderstorm Detector (BTD) was combined with thermal videography, visual imaging, standard atmospheric measurements and volcanic tremor measurements.”

Figure 1: There is a bit of confusion in this figure. White stars represent both seismic stations and BTD stations, and blue circles both villages and astronomical observatories.

Please specify that the sensor was moved from BTD1 to BTD2.

Thank you for pointing out that the symbols used were confusing. We have updated the figure and caption by appointing different symbols to the various types of sensors/localities. In addition, we now mention the time the Biral Thunderstorm Detector recorded at each location (BTD1 and BTD2). The new caption is:

Lines 72-82: “**Figure 1: Map of La Palma with the locations of Cumbre Vieja volcanic ridge and the instruments.** Google Earth satellite images (Imagery date: 7/7/2019 – newer. Data SIO, NOAA, U.S. Navy, NGA, GEBCO. <http://www.earth.google.com> [24 October 2023]) **a)** La Palma, Canary Islands (Spain), showing the location of the eruptive craters (red circle), seismic station PLPI (white triangle) and Roque de los Muchachos observatory (blue square). **b)** The 2021 Tajogaite lava flow field (red shaded area; data from the European agency Copernicus Emergency Management Service, <https://emergency.copernicus.eu/mapping/list-of-components/EMSR546> [24 October 2023]) is shown together with the location of the Biral Thunderstorm Detector (installed from 11-26 October at location BTD1 and relocated to location BTD2 on 27 October, white stars), the thermal camera (white square), the active vents (red circles) and nearby villages (blue circles).”

Lines 109-101: This positive correlation is not so evident to me. Perhaps it could be better visualized in a plume height vs. discharge rate plot.

Lines 344-347: Ok, but then you have to: 1) show better this correlation, and 2) show that plume height is only a function of mass eruption rate, independent of eruption style.

We agree with the reviewer that this positive correlation is not so evident, for which there are many arguments. The authors tried to visualise the correlation in a plume height versus discharge rate plot as the reviewer suggested, but this was unsuccessful due to difference in temporal resolution and the uncertainties in the plume height. While the electrical discharge rate can be calculated for any time window due to the continuous recording, the plume heights were only reported at regular intervals four times per day by the Toulouse VAAC and once per day by PEVOLCA. Moreover, the Toulouse VAAC reports the highest flight levels affected by volcanic ash, which is not necessarily equal to the eruption column height and could result in an overrepresentation. On the other hand, we do not take into account the effect of wind here, which could result in a strongly bend plume as we have observed during the November field campaign as well as saw in pictures taken throughout the eruption, creating a possible underrepresentation of the eruption magnitude. Even if the plume

height is accurately reported, a new pulse could have occurred at the vent without affecting the height of the already existing plume, while generating more electrical discharges as a result of enhanced plume electrification. Lastly, it was not uncommon for several vents to be active at the same time, either characterised by the same or a different explosive eruption style, which further complicates the correlation between plume height and the electrical discharge rate. The authors have made edits to the Results section in order to be more precise about the correlation between the plume height and the electrical discharge rate. Although most of the above-mentioned discussion points had already been incorporated in different sections of the manuscript, we have now combined them all together in the Discussion section.

Lines 111-118: “Electrical discharges were detected almost continuously, indicating that the eruption was very electrically active. In general, there is no clear correlation between the plume height, the electrical discharge rate and the average and maximum voltage measured by the primary antenna (Figures 2a-c). There are short periods however, e.g. 29 September – 4 November, where fluctuations in the Toulouse VAAC plume height dataset are positively correlated to the overall changes in the electrical discharge rate. In contrast, between 1 – 3 December a strong increase in the electrical discharge rate was recorded while the eruption column height decreased more than 1.5 km, indicating an anticorrelation.”

Lines 408-424: “The former fits with the observation that the fluctuations in the electrical discharge rate are sometimes for short periods of time positively correlated with changes in the plume height (e.g. 29 September – 4 November), while the latter explains why high electrical discharge rates can be detected even during times of relatively low volcanic plume height (e.g. 1 – 3 December). In general, however, no clear correlation was found between the plume height and the electrical discharge rate, which may have several reasons. It is important to take into account the difference in temporal resolution between the continuous electrical measurements and the plume heights that are reported only a few times per day. Moreover, the Toulouse VAAC focuses on the flight levels that are affected by ash, which may have resulted in an overestimation of the eruption column height, as detached ash clouds can remain at high altitudes for long periods of time. On the other hand, the plume heights have not been corrected for wind. Field campaign observations showed that the wind could strongly bend the plume, which would create an underestimate of the eruption magnitude^[7]. In addition, it is possible for new explosions pulses to eject pyroclasts into an already existing plume without affecting the height of the eruption column, while enhancing the amount of plume electrification and possibly increasing the electrical discharge rate.”

Line 112: Define acronym [PEVOLCA].

As explained in a previous comment, the authors realized that certain acronyms need to be defined earlier on in the manuscript. We have corrected this by adding a sentence at the beginning of section “Electrical activity, seismic tremor and plume height”. We highly appreciate that the reviewer paid attention to these details.

Lines 89-91: “Plume height data was obtained from two different organisations, the Toulouse Volcanic Ash Advisory Center (VAAC) and Plan de Emergencias Volcánicas de Canarias (PEVOLCA).”

Line 115: Define [PC1].

We have defined PC1 as suggested by the reviewer:

Lines 135-137: “The visual comparison of signals in Figure 2 demonstrates that the first principal component (PC1) is mostly related to changes in the absolute tremor amplitude (Figure 2e), while the second principal component (PC2) is mainly dependent on their ratio.”

Lines 122-124: To say this you should show some data on type of activity together with PC2 data, otherwise this statement is unsupported.

We have rephrased our statement and have provided a more detailed explanation regarding the volcanic tremor signals and the principal component analysis (PCA). The second principal component is mostly dependent on the ratio between the LP and VLP amplitudes. Bonadonna et al. (2022) [number 7 on the reference list] showed that the ratio between these two amplitudes reflects changes in the source mechanism of the volcanic tremor and similarly in the eruptive mechanism. To further support this statement, we have added Supplementary Figure 1 to the Supplementary Methods to show the graphical result of the PCA data decomposition. In addition, the reviewer mentioned in the review summary that we should state explicitly that there is no correlation between the volcanic tremor signals and the plume height and electrical parameters, since a negative result is still valuable information. We have added this statement to the Results section as well.

Lines 133-143: “Since the volcanic tremor amplitude is highly variable, the temporal variation of the LP and VLP components was additionally analysed using the Principal Component Analysis (PCA). The visual comparison of signals in Figure 2 demonstrates that the first principal component (PC1) is mostly related to changes in the absolute tremor amplitude (Figure 2e), while the second principal component (PC2) is mainly dependent on their ratio. Based on findings from Bonadonna et al. (2022)^[7], this suggests that PC1 is related to the intensity of the explosive activity, whereas PC2 reflects the changes in the volcanic tremor source mechanism, which is in turn connected to the type of volcanic activity. A more detailed explanation is provided in the Methods section. Also for the volcanic tremor signals and the result of the PCA applies that there is no evident correlation with the plume height and the electrical parameters.”

Lines 682-704: “In this work, we consider the volcanic tremor amplitude in the Very Long Period (VLP, 0.4-0.6 Hz) and the Long Period (LP, 1-5 Hz) frequency bands^[26]. Due to the different wavelengths of these components, they provide information about the tremor source mechanism at different depths. Using the local S-wave velocity model of D’Auria et al. (2022)^[47], we can state that the penetration depth of the VLP component inside the conduit is of few hundred metres, while that of the LP components is a few tens of metres. Therefore, the LP component is more tightly related to the explosive mechanism at the vent, while the VLP component reflects the overall amount of gas flowing through the conduit. Bonadonna et al. (2022)^[7] demonstrated that the absolute tremor amplitude is related to the intensity of the

explosive activity and the ratio between the components reflects changes in the source mechanism of the volcanic tremor and, similarly, in the eruptive mechanism. This can be explained considering that the volcanic tremor wavefield is composed dominantly of Rayleigh waves. Since the volcanic tremor amplitude is highly variable, instead of using the raw ratio, we analysed the temporal variation of these two different components using the Principal Component Analysis (PCA). Before applying PCA, we normalised the amplitudes by taking the logarithm, subtracting the average and dividing them by the standard deviation. The result of the PCA is provided in Supplementary Figure 1 in the Supplementary Methods, which shows that the temporal variation is represented by two components. The first principal component (PC1) is mostly related to the absolute amplitude of the volcanic tremor and thus reflects the intensity of the explosive activity. The second principal component (PC2) mostly depends on the ratio between the LP and VLP amplitude, indicating that it reflects the changes in the volcanic tremor source mechanism, which is in turn connected to the type of volcanic activity.”

Lines 23-28 in Supplementary Methods: “In this study, the volcanic tremor is considered within two frequency bands: Very-Long-Period (VLP 0.4-0.6 Hz) and Long-Period (LP 1-5 Hz). Using the Principal Component Analysis (PCA) on the log-normalised tremor amplitude values, the temporal variation of these two components is analysed. Supplementary Figure 1 shows the result of the PCA data decomposition. The first principal component (PC1) is mostly related to the absolute amplitude, while the second principal component (PC2) mostly depends on their ratio.”

Lines 31-34 in Supplementary Methods: “Supplementary Figure 1: Graphical representation of the PCA data decomposition. The whole dataset is represented with black circles as a function of the log-normalized VLP and LP amplitudes. The eigenvectors representative of the two PCA components are shown as red (PC1, first principal component) and blue (PC2, second principal component) arrows.”

**Figure 2a: Could be useful to add a horizontal line at vent elevation.
Please mark the time when the sensor was moved from BTD1 to BTD2.**

Adding a horizontal line at vent elevation is a great suggestion as it will make it more clear how tall the plume was above the crater rim. Note however that the elevation of the crater rim fluctuated throughout the eruption due to cycles of growth and collapse of the cone. For this reason, we have added a horizontal bar in Figure 2a, which ranges from the pre-eruption elevation (884.2 m above sea level) to the maximum elevation (1071.2 m above sea level). These values were obtained from Civico et al. (2022). In addition, we changed the y-axis of Figure 2a in order to have the crater rim at the bottom of the plot. As a result, the various datasets shown in Figure 2a became less readable. Therefore, the authors decided to plot the electrical discharge rate in a separate plot (new Figure 2c). In this plot, we also indicated the periods at which the sensor was recording at locations BTD1 and BTD2. The caption has been changed to include the new details. We thank the reviewer for the suggestions to improve this figure.

Lines 161-165: It would be very useful to have an idea of: a) how much the different signal types (especially 1 to 4) represent end-members of a continuous spectrum or, conversely, are well-defined and quantitatively distinguishable; b) how much each of the signal type is present in the whole recording.

We thank the reviewer for this comment. Signal types 1 and 4 can be considered end-members of a continuous spectrum, where the source parameters of the ash-rich explosive activity will determine the magnitude and frequency of electrical discharges that occur. Signal type 1 frequently reaches saturation levels at the primary antenna of the BTM (at a measured voltage of 0.785V), while signal type 4 barely exceeds the noise level. Signal types 2 and 3 can also be considered as part of a continuous spectrum. We believe that the duration and magnitude of these electrical bursts depend on many different factors, but it will require more investigation in the future to understand what the end-members truly are. We have added this to the Discussion section:

Lines 487-490: “Signal types 1 and 4 can be viewed as opposite end-members of a continuous spectrum covering a large range of amplitudes and frequencies for individual electrical discharges, depending on the source parameters of the ash-rich explosive events.”

Lines 578-580: “All these findings together suggest that signal types 2 and 3 are part of a continuous spectrum, where the duration and magnitude of these bursts depend on many different factors.”

Unfortunately, it is not feasible to say how much each signal type is present in the whole recording. The authors tried to determine this based on the amplitude, frequency and the duration of the electrical discharges. However, these electrical parameters vary a lot within each signal type and can also overlap with other signal types, and therefore this initial analysis was unsuccessful. After considering several options, the authors concluded that this might only be possible through machine learning. As the authors do not have machine learning experience and these signals are particularly complicated due to their high variability, this is considered beyond the scope of this study and will be investigated during a follow-up paper as the reviewer already expected. Instead, the focus of this study was to provide observational insights into the link between the electrical signals and the explosive eruption style, which has not been done before. Nonetheless, the authors agree with the reviewer that more information about the variation in the signals throughout the time of monitoring would be useful for the reader. For this reason, the authors have decided to include a panel that shows the average and maximum values of the absolute voltage per hour measured by the primary antenna for the whole recording (new Figure 2b). This shows how frequently the electrical activity changes in amplitude. In addition to the electrical discharge rate, we have also added a normalised discharge rate (Figure 2c), which is the electrical discharge rate times the normalised maximum voltage per hour. This shows the relationship between the electrical discharge rate and the amplitude of the discharges. We have included these results in the text:

Lines 91-95: “These two datasets are compared with the time series of atmospheric temperature at the 0°C, -10°C and -20°C isotherms (Figure 2a), the average and

maximum value of the absolute voltage (V) per hour (Figure 2b) and the electrical discharge rate (discharges/hour) as detected by the BTD and identified by the volcanic lightning detection algorithm during the observation period (Figure 2c).”

Lines 123-127: “The normalised discharge rate, which is the electrical discharge rate times the normalised maximum voltage (see Methods section), demonstrates that there is no relationship between these two parameters (Figures 2b-c). The average and maximum voltage fluctuate strongly throughout the eruption, with the primary antenna repeatedly reaching the saturation level of 0.785V (Figure 2b).”

Lines 152-159, Caption of Figure 2: “**b**) The one-hour average (black line, left y-axis) and maximum (purple line, right y-axis) of the absolute voltage measured by the primary antenna of the BTD. **c**) The electrical discharge rate (light green vertical bars) and the normalised discharge rate (dark green vertical bars) per hour. The normalised discharge rate is the electrical discharge rate times the normalised maximum voltage measured per hour by the primary antenna. The light and dark red vertical bars show the electrical and normalised discharge rate, respectively, measured during a nearby thunderstorm. The arrows indicate the time of monitoring at each location (BTD1 and BTD2).”

Lines 403-406: “This is corroborated by the fact that the highest discharge rates (>5000 discharges/hour) were detected during various explosive eruption styles and there is no correlation between the measured voltage and the electrical discharge rate, as is evident from the normalised discharge rate (Figures 2b-c).”

Lines 631-638: “In addition, the average and maximum value of the absolute voltage (V) measured by the primary antenna of the BTD were calculated per hour. To show the relationship between the electrical discharge rate and the amplitude of the discharges, we normalised the electrical discharge rate by normalising the maximum measured voltage by the saturation level of the primary antenna (0.785V) and multiplying this with the electrical discharge rate. A 1:1 ratio between the normalised and calculated electrical discharge rate indicates that the largest electrical discharge saturated the primary antenna, while a small ratio indicates that the largest discharge within that hour was relatively low in amplitude.”

Figure 4b: Due to the very different y-axis scale values, it is difficult to compare this signal with the snapshots of the same signal provided in figure S3. Perhaps using a log scale on the y-axis could help visualising both the small- and large-scale parts of the signal.

We agree with the reviewer that the y-axis scale of Figure 4b makes it difficult to compare the various electrical signal types due to their large range in amplitudes. Unfortunately, a normal log scale cannot be used, since it cannot be applied to negative values. The authors considered using a fake log scale for the negative values but decided against it in the end, as it does not represent the data correctly. Ultimately, we decided to leave Figure 4b as it is, as the authors believe it is important to show the entire range in electrical activity. Instead, we have plotted the maximum value of the absolute voltage per 5 minutes of data on a log-scale and have added this to the electrical discharge rate data in Figure 4c. This helps visualising both the low- and high-voltage electrical signals, but you lose the information on the polarity of the electrical discharges. In addition, we have added another plot to

Supplementary Figure 3 (new Supplementary Figure 3a). Here we show the same data as in Figure 4b, but using a smaller y-axis scale $[-0.1, 0.1]$ instead of $[-0.8, 0.8]$ in order to make the low-voltage electrical discharges better visible. Here we have also indicated the time windows shown in Supplementary Figures 3b-i using blue vertical bars to help the reader recognize when each signal occurred during the case study. We hope the reviewer understands our reasoning and agrees with our solution.

Figure 4c: At this time there is no change in discharge rate despite changes in type of signal and eruption style. Did you try to use a different metric for the electrical signal? Why not use one that beside rate of the discharges also includes their amplitude (e.g., frequency times mean amplitude in a time window)?

We thank the reviewer for this comment, it is an excellent suggestion. As mentioned in the previous comment, we have plotted the maximum value of the absolute voltage per 5 minutes of data and added it to the electrical discharge rate data in Figure 4c. Plotting the frequency times the mean amplitude, as the reviewer suggested, would have been effective as well, but the authors decided against it in the end because showing both parameters separately provides more information.

Upon re-examining the thermal data, the authors believe that the change in eruption style might have occurred slightly earlier (around 18:22 instead of 18:55 UTC). During this half hour, the vent area and the eruption column are most of the time obscured by clouds, which made it very difficult to determine the exact time of the change in explosive activity. This change corresponds very well, however, to a decrease in the electrical discharge rate and the maximum value of the absolute voltage, which could suggest that the explosive activity was starting to change. From 18:46 UTC on, the thermal images (albeit out of focus) showed that the activity had indeed changed to lava fountaining. We have added this updated information to the Results section:

Lines 281-284, Caption of Figure 4c: “c) Electrical discharge rate (discharges/5 minutes) as green vertical bars (left y-axis) and the maximum value of the absolute voltage measured by the primary antenna per 5 minutes in purple (right y-axis with log-scale).”

Lines 290-298: “Around 18:22 UTC, the electrical discharge rate and the maximum value of the absolute voltage decreased (Figure 4c). This change was accompanied by a brief increase in the seismic tremor amplitudes and the value of PC1, as well as a temporary decrease in the value of PC2 (Figures 4d-f). During this time, the thermal data showed that the vent area was obscured by clouds most of the time, and therefore no information is available on the explosive activity. It can be speculated, however, based on the measurements that the activity was starting to change at this point. Around 18:46 UTC, once the clouds had cleared away, the thermal images showed that the explosive activity had indeed changed to a lava fountaining phase again (Figure 4a-5), dominated by the ejection of incandescent bombs.”

Figure 4e: No obvious change in tremor despite change in activity [at 03:00 UTC]. No obvious change in tremor despite change in activity [at 13:09 UTC].

We thank the reviewer for these comments, but the authors disagree with the statements. Just before 03:00 UTC, the tremor amplitude shows a highly variable amplitude, while PC2 shows a clear descending trend. Before 13:09 UTC, the VLP component shows a “bump” which is also reflected by both PC1 and PC2. We have rephrased already present information on the changes around 03:00 UTC and added information on the changes around 13:09 UTC to point this out more clearly:

Lines 225-227: “These changes are accompanied by a high variability in the tremor amplitudes and the value of PC1, while the value of PC2 shows a descending trend (Figure 2d-f).”

Lines 236-237: “This change was preceded by a temporary increase in both the tremor amplitudes and the value of PC1 and a temporary decrease in the value of PC2.”

Lines 225-226: Are you sure there was no ash plume? By night it is hard to assess ash content. Perhaps you can add some other information supporting this assessment.

The reviewer asks a valid question here. The authors do not claim that there was no ash present at all. What we would like to emphasize here is that in contrast to the ash-rich lava fountaining on the evening of 3 November, a well-developed ash plume of several kilometres in height remained absent during the lava fountaining phase on the evening of 4 November. It is true that at night it is hard to assess the actual ash content that is ejected, but whether or not such a large ash plume was produced, could still be visibly observed by us. However, we can further support this statement by including thermal images of the activity in the Supplementary Material (new Supplementary Figure 5), which clearly show a difference in the amount of ash that is being emitted. The activity on the evening of 3 November produced relatively dense, turbulent eddies within the eruption column which had an initial diameter of 300-500 m, while the eruption columns produced by the activity on the evening of 4 November had an initial diameter of 100-200 m and appeared less dense. We agree with the reviewer that additional information is useful here to support our statement regarding the ash content of the different activities, so we have added the above information to the text:

Lines 217-219: “The thermal videography showed an initial diameter of 300-500 m of the eruption column with relatively dense, turbulent eddies (Supplementary Figure 5a).”

Lines 301-302: “As a result, the eruption column only had an initial diameter of 100-200 m and dense, turbulent eddies remained absent (Supplementary Figure 5b).”

Lines 66-71 of Supplementary Material: “Supplementary Figure 5 displays the difference between two phases of lava fountaining observed during the study case of 3-4 November. Lava fountaining on the evening of 3 November generated a relatively dense, turbulent 3-km tall ash plume. The initial diameter of the eruption column varied between 300-500 m. In comparison, a lava fountaining phase on the evening of 4 November emitted less ash and did therefore not form such a dense ash plume. Instead, the initial diameter of the eruption column was smaller (100-200 m).”

Lines 92-97: “Supplementary Figure 5: **Thermal infrared frames showing two different phases of lava fountaining.** a) Ash-rich lava fountaining at the main vent producing relatively dense, turbulent eddies and a 3-km tall ash plume at 23:44:55 UTC on 3 November 2021. A second vent is currently active on the lower right. b) Lava fountaining at 23:48:13 UTC on 4 November 2021, emitting less ash in comparison to the explosive activity on the evening of 3 November shown in panel a. As a result, no dense, turbulent ash plume was formed.”

Subsection “Linking electrical activity to explosive eruption styles”: This narrative description of shifting volcanic activity is interesting but would be much more robust if supported by other data. If your thermal imaging covers the whole day of Nov. 4, a figure like Fig. 5 would substantiate the correlation between changing activity and changing electrical signal.

This is something that the authors had tried before submission, but is unfortunately not feasible for several reasons: 1) There are several data gaps; 2) Clouds obscured the vent area partially or completely for periods of time and thus no information on the explosive activity could be extracted; 3) Due to the changing weather conditions, the thermal camera went out of focus at times, which decreases the quality of the rise diagram; 4) Plotting 27 hours in one rise diagram is only feasible if a frame is taken every ~10 minutes (in comparison to every 2 seconds for a 5-minute time window), meaning that a lot of information is lost. We do appreciate the suggestion a lot, as it could have been a great way to show the changing activity with time.

Figure 6a-b: Instead of pictures, a rise diagram of the thermal imagery in parallel with the electrical signal would be much more effective in showing the link between activity and electrical signal.

Figure 6c: Transition between bursts and single discharges (types 2 and 1)?

Many thanks to the reviewer for this comment and question. We agree with the reviewer that a thermal rise diagram will make it easier to investigate the link between the electrical signals and the Strombolian activity. Unfortunately, we do not have thermal data for the entire period of the electrical signals shown in original Figure 6c (19:00-19:30 UTC on 6 November 2021). For this reason, we have changed the figure and now show a later period during the same phase of explosive activity (01:20-01:40 UTC on 7 November 2021) for which we do have thermal data available (new Figure 6d). As the rise diagram is generated over a longer time period, we applied the algorithm of Gaudin et al. (2017a,b; numbers 45 and 46 on the reference list) to every 120th thermal frame (i.e. every 8 seconds of recording) instead of every 30th thermal frame.

To answer the question of the reviewer, it is indeed true that at times single discharges were detected in between bursts of electrical discharges. However, the overall electrical activity can be described as a mixture of signal types 2 and 3 as these clearly dominate. Comparing the electrical signals to the rise diagram, no clear correlation can be found. To provide further details, the authors have decided to replace the previous thermal image and photograph (original Figures 6a and 6b) with three sequences of thermal images (with 5 seconds between frames) that correspond to different electrical signals (little to no electrical activity, minutes-long bursts and seconds-long bursts; new Figures 6a-c). Although small differences can be observed, such as in the height of the eruption column, no immediate explanation can be provided for the variation in the electrical signal. More detailed time series of e.g.

eruption column height, particle ejection velocity, mass eruption rate and particle size distribution could help us understand the underlying cause for fluctuations in the electrical activity better in the future.

We have updated Figure 6 as well as its caption. In addition, we have made the necessary edits in the text to include these changes:

Lines 333-347: “This type of activity predominantly produced a mixture of seconds-long and minutes-long bursts of electrical activity with measured voltages dominantly between 0.001-0.0025V (signal types 2 and 3, Figure 6d), although at times single electrical discharges of similar amplitude occurred in between bursts. Comparing three sequences of thermal frames, corresponding to different electrical signals, demonstrates that there are small differences in the explosive activity on short timescales, especially in the size of the eruption column. Very minor Strombolian activity, reaching heights up to 200 m a.c.r. (Figure 6a), produced little to no electrical activity. A burst of quasi-continuous electrical discharges lasting ~2.5 minutes was detected during a period where pyroclasts were ejected up to 400 m a.c.r. (Figure 6b), while a shorter burst lasting approximately 20 seconds was generated during intermediate Strombolian activity reaching heights of 100-300 m a.c.r. (Figure 6c). Nonetheless, no obvious correlation between the electrical signals and the rise diagram can be observed that could explain the variation in the duration of the bursts (Figure 6d). Although there are periods where (longer) bursts of electrical activity coincide with pyroclasts being ejected to greater heights, at other times the opposite seems true.”

Lines 349-357: “Figure 6: **Strombolian activity producing incandescent bombs and a small semi-opaque ash plume on 7 November.** Single frames extracted every 5 seconds from a thermal video taken at 15 fps, with the first frame taken at **a)** 01:29:00 UTC, **b)** 01:31:00 UTC and **c)** 01:38:00 UTC on 7 November 2021. **d)** Thermal infrared rise diagram showing the evolution of the maximum temperature (blue is cold and yellow is hot) as a function of time and height (left y-axis). The crater rim of the main active vent is at a height of 0 m in the diagram. In addition, the voltage (V) measured by the primary antenna of BTD2 (black line, right y-axis) is shown, predominantly displaying a mixture of signal types 2 and 3. The red rectangles indicate the time periods of each sequence of thermal frames shown in panels **a-c.**”

Lines 557-560: “In addition, small variations in the size of the eruption column, as observed in the thermal data (Figure 6), may further affect the duration, although this relationship is not always evident and requires further investigation.”

Lines 670-672: “This analysis was carried out for every 30th thermal frame for a 5-minute time window and every 120th frame for a 20-minute time window (i.e. every 2 and 8 seconds of recording, respectively).”

Line 336: Charging of particles at the vent, not charging of the vent itself, I guess.

Thank you, the reviewer is correct about this. We have corrected the sentence, it now reads:

Lines 399-401: “Behnke et al. (2014)^[32] proposed that fluctuations in the electrical discharge rate throughout a single eruption can be either caused by intensified charging of particles at the vent due to an increase in source flux or enhanced plume electrification due to ice nucleation.”

Lines 342-343: But in general and also specifically at Tajogaite mass eruption rate is NOT independent of eruption style. In addition you only assessed eruption style, not mass eruption rate during your measurements. How do you support this statement?

Lines 372-374: You could be more specific on mass eruption rate and exit velocity by citing the values for eruption styles of the Tajogaite eruption presented in Taddeucci et al., 2023. Although activity style names there may be different from other papers you cite, it shouldn't be difficult to reconcile them by using the description of the activity itself.

Lines 402-403: Again, you could provide some number here.

Many thanks for this suggestion. We agree that discussing the mass fluxes and maximum particle ejection velocities estimated in Taddeucci et al. (2023) [number 9 on the reference list] will provide more insight into the differences between the various explosive eruption styles and the electrical signals that they generated. In particular, Behnke et al. (2014) [number 32 on the reference list] suggested that mass fluxes throughout a single eruption can affect the electrical discharge rate by enhancing/diminishing the charging of particles at the vent. Regarding the differences in terminology, we believe that the following explosive eruption styles can be considered similar: strong ash emissions = “ash-rich jets”, mild ash emissions = “column and plume”, Strombolian activity = “spattering”, gas jetting = “ash-poor jets” and lava fountaining is named the same. Mass fluxes were not estimated for strong and mild ash emissions, so we will only compare these values for gas jetting, Strombolian activity and lava fountaining. Below you find the edits that we have made in the Discussion section.

Lines 524-528: “Taddeucci et al. (2023)^[Error! Reference source not found.] found that although all explosive activity was pulsating, gas jetting (named “ash-poor jets” in the study), Strombolian activity (“spattering”) and lava fountains had shorter pulse intervals than the strong ash emissions (“ash-rich jets”). Also the maximum particle ejection velocity (MPEV) was higher for these three explosion styles in comparison to strong ash emissions^[9Error! Reference source not found.].”

Lines 544-564: “The lava fountains were driven by the highest kinetic energy and were estimated to have the greatest mass flux, ranging between $0.8 - 2.8 \times 10^4$ kg/s^[9], which was reflected in their relatively high eruption columns^[41,42]. A high mass flux and MPEV (~24-100 m/s) provide favourable conditions to rapidly generate a lot of plume electrification through a large number of particles fracturing and colliding at a fast rate, which could explain the overall long duration of these bursts. Short term variations most likely affected the efficiency of charging and discharging^[32], which could be the reason for shorter bursts interrupting the predominantly minutes-long bursts of electrical activity. In contrast, Strombolian activity was found to have both a relatively intermediate mass flux of $4 - 9 \times 10^3$ kg/s and an intermediate MPEV of

~26-32 m/s^[9]. The same study also showed that this type of activity consisted for ~18-31% of bombs with a diameter >0.5 m. The presence of fine particles is key for the generation of charge^[14,15]. Hence, fluctuations in the grain size distribution, such as the proportion of large bombs, could have either hindered or promoted the plume electrification processes and thus affecting the duration of the electrical bursts. In addition, small variations in the size of the eruption column, as observed in the thermal data (Figure 6), may further affect the duration, although this relationship is not always evident and requires further investigation. The mass flux of gas jetting was estimated to be the lowest of these three eruption styles, ranging between $0.2 - 8 \times 10^3$ kg/s^[9], creating a relatively low density eruption column as particles would reach similar heights as in lava fountains. A lower density would mean less particles colliding and therefore less charge creation, which could explain why gas jetting generated the shortest bursts of electrical activity.”

Lines 350-352: I agree it is a complicated situation, but your fantastic data, almost continuously covering a long time period, pose you in the best possible position to unravel it. This is why I suggest to tighten electrical signals to shifting activity in the most robust possible way, going beyond the 3-days examples you provide.

We are happy that the reviewer is enthusiastic about the results and sees the value of this dataset. Unfortunately, it is not feasible to tighten the electrical signals to shifting activity for the entire time of monitoring. As mentioned during a previous comment, machine learning will be necessary to automatically determine the signal types for the entire period. This will most likely take months of research and analysis as the authors do not have experience with machine learning and the signals are very complex due to their high variability. Moreover, although there are abundant photos and videos available of the eruption to provide information on the explosive activity, the authors found that both the temporal and visual resolution are often insufficient to provide the necessary detailed information, like we obtained for the case study of 3-4 November. Despite of this, the authors agree with the reviewer that this would be very valuable to try in the future and are aiming to do this for a follow-up paper.

Lines 353-354: Is it always true that discharge rate is highest phases 2 and 3? What phase dominates during other periods of high discharge rate (e.g., beginning of December)?

We are really happy the reviewer asked these questions, as we realised upon revising Figure 2 that the highest electrical discharge rate had shifted when calculating the number of discharges per hour instead of per 6 hours. This taught us that it is best not to make any statements about when the highest electrical discharge rate occurred, but instead to focus on what kind of explosive eruption styles typically produced high discharge rates, as the reviewer suggested. In addition, one cannot compare the electrical discharge rates detected by BTD1 with those detected by BTD2, as BTD2 had a higher detection efficiency due to its closer distance to the active vents. As a result, the electrical discharges were detected with a higher amplitude at BTD2 and therefore the volcanic lightning detection algorithm was likely more successful in identifying signals as volcanic lightning.

There were several moments where the electrical discharge rate exceeded 5000 discharges/hour, as recorded by BTD2 and identified by the detection algorithm. Through direct observations during our field campaign in November 2021 and

through the use of videos posted by Instituto Volcanológico de Canarias on X (Twitter), we obtained information on the explosive eruption styles that corresponded to these high discharge rates:

Lines 118-123: “The BTD recorded an electrical discharge rate >5000 discharges/hour at various occasions throughout the time of monitoring. These discharge rates were detected during different eruption styles, including strong ash emissions, ash-rich lava fountaining and intense Strombolian activity, as was observed both during a field campaign early November 2021 and through videos posted by Instituto Volcanológico de Canarias on X (Twitter).”

Lines 403-406: “This is corroborated by the fact that the highest discharge rates (>5000 discharges/hour) were detected during various eruption styles and there is no correlation between the measured voltage and the electrical discharge rate, as is evident from the normalised discharge rate (Figures 2b-c).”

Line 372: Which style: strong ash emission or lava fountaining? Or both? But then, what is the difference between the two?

We thank the reviewer for asking these questions. The authors have clarified that they are talking about both the strong ash emission as well as the lava fountaining. Although there are differences between the two explosive eruption styles regarding fragmentation mechanism, eruption dynamics and source parameters, they both produce electrical activity characterised by individual high-amplitude electrical discharges (signal type 1). It is what they have in common, namely the formation of a well-developed volcanic ash plume due to the ejection of large amounts of ash, that forms the basis for this type of electrical signal. Ash is the main carrier of charge. The fragmentation and collision of ash particles creates a lot of charge, and the subsequent charge separation and formation of charge clusters during rise and convection of the plume allows the generation of volcanic lightning of high measured voltage. The other eruption styles observed during the 2021 Tajogaite eruption lacked the formation of a well-developed volcanic ash plume and therefore did not produce electrical signals of type 1. In the case of mild ash emission, the charging is less intense and therefore the electrical discharges that are generated are of much lower measured voltage, but the concept remains the same as for the strong ash emission. The authors have realised that it is important to emphasize more that it is the formation of a large volcanic ash plume that forms the foundation of this type of electrical activity. Below you find the revised paragraph:

Lines 442-452: “Although there are differences between these two explosive eruption styles, both ejected a large amount of ash for hours without interruption, which is responsible for generating substantial charge in the plume through the fracturing and collisions of particles. The development of a tall volcanic ash plume, especially one that is sustained for several hours, allows the build-up of a strong electric field through charge separation and the formation of charge clusters^[14,33,34], generating volcanic lightning of high measured voltage. Other explosive activity during the Tajogaite eruption, such as gas jetting, Strombolian activity or lava fountaining ejecting significantly less ash, lacked the formation of such a well-developed ash plume. This shows that the ejection of large quantities of ash, subsequently undergoing convection and rising to form a mature plume, forms the foundation for the generation of type 1 electrical signals.”

Lines 382-383: Could this be related to turbulence within the plume?

We thank the reviewer for asking this question. The slow-varying electrical signal is not directly the result of turbulence, but is instead caused by the movement of charge relative to the BTD (meaning charge moving towards and/or away from the sensor). This can be particularly strong for example when the plume is moving overhead. Nonetheless, turbulence does play a role in charging by promoting particle collisions and will influence the charge distribution, so it is part of the equation, but not the cause for the slow-varying electrical signal.

Lines 386-388: Any reference or data to support this interpretation?

Lines 389-391: This part seems largely speculative, with assumptions on the type of ash being erupted, on changes in conduit diameter, and on the location and source mechanism of tremor.

We thank the reviewer for these comments. Although the scenario described here is merely a speculation, we agree that additional data could help support this. Bonadonna et al. (2022) [number 7 on the reference list] demonstrated the relationship between the stratigraphy, which is directly related to the eruptive style, and the variations in the volcanic tremor both in terms of amplitude and ratio, which are reflected in PC1 and PC2, respectively. Between 1-3 November, a clear shift in the stratigraphy was observed (from Middle Unit MU2-5 to MU6, following Bonadonna et al., 2022), which was accompanied by a clear decrease in the VLP amplitude and increase in the LP/VLP ratio. A similar but slightly smaller decrease in the VLP (and LP) amplitude and increase in PC2 is observed on 4 November. Moreover, in addition to juvenile clasts, Middle Unit MU6 contains oxidised red, dull-looking scoriae clasts, which could indicate recycled material. Although the volcanic tremor data does fit this scenario, it is still largely a speculation, which we should state more clearly in addition to providing the above data:

Lines 461-464: “We speculate that this swift change in volcanic activity observed on 4 November was the result of a partial collapse of the shallow plumbing system, which would cause recycled ash to be ejected in addition to juvenile material.”

Lines 467-474: “Bonadonna et al. (2022)^[7] demonstrated the relationship between the stratigraphy, which is directly related to the eruption style, and the variations in volcanic tremor both in terms of amplitude and ratio, which are reflected in PC1 and PC2, respectively. Between 1-3 November, a clear shift in the stratigraphy (from Middle Unit MU2-5 to MU6) was accompanied by a similar, slightly larger decrease in the VLP amplitude and increase in LP/VLP ratio^[7] in comparison to the event on 4 November. Moreover, besides juvenile clasts, Middle Unit MU6 contains oxidised red, dull-looking scoriae clasts^[7], which could indeed be an indication of recycled material.”

Lines 477-481: “Hence, if our speculation is correct, the occurrence of volcanic lightning during this event shows that instability of the upper volcanic conduit and the crater walls can generate adequate charging by remobilisation and recycling of older incoherent material in addition to that provoked by the ejection of juvenile tephra.”

Line 410: Electrical signals of type 1, 2 and 3 are all present during lava fountains, suggesting this activity style covers a very broad range of eruption dynamics.

This is an excellent point that needs to be mentioned in the text. It is true that lava fountaining generates electrical signals of type 1, 2 and 3, which would suggest that it covers a wide range of eruption dynamics as well as source parameters. The main difference between the generation of signal type 1 versus types 2 and 3, is the amount of ash that is being ejected and whether this results in a well-developed volcanic ash plume. The difference between signals 2 and 3 is more complex, as is shown by the fact that they can occur interchangeably, and probably comes down to short term variations in the eruption dynamics and source parameters. This remains to be investigated in more detail during similar eruptions in the future using instruments that can quantify these changes, but is beyond the scope of this study. We thank the reviewer for this comment and have included the statement together with the additional explanation in the text:

Lines 531-539: “Interestingly though, the case study shows that lava fountains can generate electrical signals of type 1 as well as types 2 and 3, suggesting that this activity style covers a wide range of eruption dynamics and source parameters. The key difference that determines what type of electrical signal is generated during lava fountaining (type 1 versus types 2 and 3), is the amount of volcanic ash that is being ejected and whether this results in a well-developed plume or not (Supplementary Figure 5). The difference between signal types 2 and 3 seems more complex, as they share similar characteristics and can occur interchangeably on short time scales, but quantifying this will require detailed time series of the source parameters during future eruptions.”

Lines 429-431: Lava fountains with short-duration thermals and no convection? But at this point what is the difference between gas jetting, Strombolian activity and lava fountains?

Thank you for asking these important questions. We have realised that we did not explain this entirely correctly and apologise for this. The thermals are of short duration in the sense that the eruption column is limited to the gas-thrust phase and is short-lived as a result of a large proportion of ejected particles quickly falling back down again instead of continuing to rise. Our statement that gas jetting, Strombolian activity and lava fountains do not develop strong convection is, however, not correct. What we meant to say instead is that this type of explosive activity does not develop a large convecting ash plume. As the reviewer has pointed out in the next comment, all explosive activity during the 2021 Tajogaite eruption was pulsating, but the pulse intervals were shorter for these three explosive eruption styles in comparison to the strong ash emissions. It is important to point out the short-lived eruption columns, the particle ejection velocity and the quickly pulsating nature of these explosions, as they most likely created the dynamics that were needed to generate these bursts of electrical discharges. We have rephrased and clarified the paragraph:

Lines 515-531: “The thermals associated with these explosive styles were of short duration so that the eruptive column was mainly limited to the gas-thrust phase, resulting in eruption column heights of only a few hundred metres up to 1 km a.c.r. depending on the eruption style. Moreover, the eruption columns were short-lived due to a large proportion of ejected particles falling back down close to the vent^[40Error! Reference source not found.], although overlap of falling and rising pyroclasts ejected during the next pulse occurred regularly^[9Error! Reference source not found.]. Consequently, these types of explosive eruption styles did not build up strongly electrified, convecting plumes with large charge clusters, and as a result did not generate the conventional volcanic lightning as in the case of sustained ash-rich eruptive episodes. Taddeucci et al. (2023)^[9] found that although all explosive activity was pulsating, gas jetting (named “ash-poor jets” in the study), Strombolian activity (“spattering”) and lava fountains had shorter pulse intervals than the strong ash emissions (“ash-rich jets”). Also the maximum exit velocity of the particles was higher for these three explosion styles in comparison to the strong ash emissions^[9]. So rather, the short-lived eruption columns but quickly pulsating nature of these explosions likely resulted in faster charging and more efficient discharging due to increased turbulence, explaining these bursts of low-amplitude electrical discharges.”

Lines 435-437: Taddeucci et al., 2023 Fig. 9 shows that all the activity at Tajogaite was pulsating, and that there is a direct correlation between pulse frequency and pyroclast exit velocity, so it may be hard to separate the effect of the two.

The reviewer is correct here and we are grateful to be reminded of this fact. As we have explained in the previous comment, it most likely was a combination of short-lived eruption columns and relatively short pulse intervals that formed the right conditions to generate this particular electrical activity. We have corrected our sentence and included the findings of Taddeucci et al. (2023) [number 9 on the reference list] regarding the pulse intervals for the various explosive styles during this eruption. Please find our revised text in the previous comment.

Lines 442-443: At line 432 you state that in all styles eruption columns are limited to few hundred meters, while here you claim different column heights.

We thank the reviewer for pointing out that this requires clarification. The thermal images taken on 4 November 2021 showed that the eruption columns generally reached heights up to 1000 m a.c.r. during gas jetting (although the hot core of the erupted mass was typically limited to lower heights), up to 500 m a.c.r. during Strombolian activity and 5000-1000 m a.c.r. during lava fountaining. Note that pyroclasts might have continued to rise due to their upward momentum, depending on wind conditions. We realised that we had not mentioned the overall eruption column heights for the phases of gas jetting that were observed on 4 November, so we have added two sentences regarding that in the Results section. In addition, we have rephrased and clarified the sentence that the eruption columns were limited to a few hundred meters for all eruption styles, as this is not correct. In the end, further comments of the reviewer have helped us see that it is also important to focus on the differences in mass eruption rate and particle ejection velocity and not only the height of the eruption columns, so we changed our discussion as was mentioned in a previous comment.

Lines 241-243: “From the thermal images can be deduced that cooling of the pyroclasts is of minor importance in the first 300 m of vertical transport and that the pyroclasts continued to rise up to 1000 m above the crater rim (a.c.r.).”

Lines 250-252: “During this time, the thermal images show an increment in the intensity of the gas jetting, evident from the increase in the total amount of hot material that is ejected up to greater heights (up to 700 m a.c.r.).”

Lines 515-518: “The thermals associated with these explosive styles were of short duration so that the eruptive column was mainly limited to the gas-thrust phase, resulting in eruption column heights of only a few hundred metres up to 1 km a.c.r. depending on the eruption style.”

Lines 443-445: I guess you are referring to large, bomb-sized pyroclasts in all three cases. This is a very interesting information. Please add the relative data in the supplementary information.

Based on this comment, we have realised that we should have been a bit more specific here. Generally, the core temperature of the lava fountain or Strombolian activity was relatively higher than during gas jetting. Unfortunately, we are unable to provide absolute values of the temperature due to calibration problems. The thermal frames will be publicly available at a one-minute time interval. The data can be reviewed using the temporary link provided at the beginning of this document. We have corrected the sentence as follows:

Lines 564-566: “Moreover, the thermal images showed that the core temperature during phases of Strombolian activity and lava fountaining was often higher in comparison to the material ejected during gas jetting.”

Lines 453-455: How can you tell that in this period of gas jetting intensity and temperature increased? Consider that the average temperature of the erupted mass (gas, ash, lapilli, bombs) may be difficult to retrieve from thermal imagery.

We thank the reviewer for this comment, because this is indeed true. There is a clear relative increase in the core temperature of the erupted mass at 15:10 UTC compared to the very first stages (between 13:09-13:39 UTC) of the gas jetting phase, but compared to minutes before, the change in temperature is minor and cannot be determined properly. Instead, what we should mention here is that it is the total amount of ejected hot material increases together with the intensity. We have corrected this in both the Results and Discussion section:

Lines 250-252: “During this time, the thermal images show an increment in the intensity of the gas jetting, evident from the increase in height of the hot core of the eruption column (up to 700 m a.c.r.) and the total amount of hot material that is ejected.”

Lines 574-576: “Similarly, a short period of an increased amount of hot material being ejected up to greater heights during gas jetting resulted in longer bursts (comparing Supplementary Figure 4e to 4f).”

Lines 475-478: Could you comment on if and how the different distance from the vent may affect the signals?

We are grateful to the reviewer for asking this question, as it is important information that the authors forgot to include in the manuscript. The electric field decreases proportional to the distance cubed at frequencies <100 Hz, like in the case of the BTM. This means that BTM2, which was installed 880 metres closer to the active vents, recorded the electrical signals with a higher detection efficiency than BTM1. As a result, the electrical signals detected by BTM2 have a higher amplitude, and thus generally also a higher signal-to-noise ratio, which facilitates the identification of electrical discharges through the volcanic lightning detection algorithm. For this reason, one cannot directly compare the electrical discharge rate detected at BTM2 with that detected at BTM1. We have added this information to the Methods and Results section and have added two new references (Wilson, 1921; Bennett & Harrison, 2013) to the list to support the information on the changes in the electric field with distance:

Lines 95-99: “It is important to take into account that the detection efficiency differs between the two BTM locations (BTM1 and BTM2) as the electric field decreases with the distance cubed at frequencies <100 Hz^[27,28]. BTM2 generally had a higher detection efficiency due to its closer location to the active vents. This affects the electrical parameters calculated from the measurements.”

Lines 185-187: “The ranges in measured voltage for signal types 1-4 are based on the detections of BTM2, which are generally of higher amplitude compared to BTM1 due to its closer location to the active vents.”

Lines 639-645: “Although the electrical parameters are calculated and provided for both BTM locations, these cannot be compared to each other. At frequencies <100 Hz, the electric field decreases proportional to the distance cubed^[27,28]. BTM2 had a higher detection efficiency due to it being installed 880 m closer to the active vents. As a result, the electrical signals detected by BTM2 typically have a higher amplitude, and thus generally also a higher signal-to-noise ratio, which facilitated the identification of electrical discharges by the algorithm. For this reason, this study focuses mainly on the measurements of BTM2.”

Lines 523-525: But you show only one single rise diagram and none of these data!!!

The reviewer has a valid point here. We have included another rise diagram in our manuscript (Figure 6d) to better show the variations in electrical signals and explosive activity together. As explained in a previous comment, it was unfortunately not feasible to provide a rise diagram that covers the entire case study of 3-4 November. The thermal camera did not record continuously for the entire period and went out of focus for periods of time. In addition, clouds obscured the vent area at times and thus no information is available on the explosive activity. Moreover, a lot of information gets lost when plotting such an extensive time window (27 hours) in one diagram. It is true that we do not show any of the data that can be extracted from the rise diagram as we mentioned in the Methods section. We use the rise diagrams to observe changes in the explosive activity (such as Figure 5c) and to see if there is

any relation between the explosive activity and the electrical signals that were detected. We agree that we should say this more clearly in the Methods section, so we have rephrased the sentence:

Lines 673-676: “In this study, the rise diagrams are used to investigate the link between the electrical signals and the pulsating explosive activity and changes thereof. More detailed future analysis could provide information on the erupted products (ash- or bomb-dominated) and the rise velocity (based on the slope of the traces) as well^[45,46].”

The reviewers' comments are given in **bold** (with line numbers corresponding to the original draft), author replies in *italics* and manuscript edits in quotation marks and regular text, with corresponding line numbers.

Note that the DOI provided in the Data Availability statement has not been registered yet. All data will be made public upon acceptance of the manuscript. The data is currently available for review with the following temporary link:

<https://dataservices.gfz-potsdam.de/panmetaworks/review/5c0d2f8e3859809de54e1ca112f47558dffc420d571ca339cc6eea543f1f0107/>

Reviewer 2

This paper presents analysis of new observations of electrical discharges during a recent volcanic event in the Canary Islands. Analysis of concurrent datasets (electrical, seismic, thermal imaging, plume height data) has been carried out to explore the use of electrical measurement as a volcano monitoring technique. This paper provides novel illustration of the potential for electrical monitoring as an eruption monitoring technique, and will be of interest to others in the community and will influence decisions over observations to be made at future eruptions.

We are very thankful to the reviewer for the clear and kind comments that helped us improve the paper by correcting the flow of the text and clarifying certain parts through either rephrasing or adding information. Thank you for taking the time to review our manuscript.

Specific points:

1. I would have appreciated a signpost to the 'Methods' section at the beginning of the 'Results' section (i.e. something along the lines of 'Details of the measurement set ups are given in the Methods section) as some acronyms were not defined on first use but instead were subsequently explained in the Methods section. For example, line 260 could do with a reference to Vossen et al and a cross reference to the Methods section; LP and VLP mentioned in line 116 and defined in lines 532&534.

Many thanks to the reviewer for pointing this out. The authors apologize for this. The manuscript was originally written with the Methods section coming after the Introduction section. We have realised that we forgot to correct the location of a few of the acronym definitions in the text upon rearranging the order of the sections. In addition, as the reviewer suggests, we understand it is important to make reference to the Methods section to clarify that specific information on the setup can be found there. In addition to a few other acronyms that we forgot to define earlier on in the manuscript, we have edited the text in the following places as suggested by the reviewer:

Line 88-89: “Details of the monitoring setup, data collection^[26] and analysis are provided in the Methods section.”

Lines 128-129: “Both the Very Long Period (VLP, 0.4-0.6 Hz) and the Long Period (LP, 1-5 Hz) tremor amplitude varied throughout the eruption (Figure 2d).”

Lines 306-309: “However, contrary to the signal shown in Figure 3b-c, these signals did not have only positive covariance values, which is one of the criteria of the volcanic lightning detection algorithm (see Methods section and Vossen et al. (2021)^[24Error! Reference source not found.] for more information).”

2. Can the authors say more about why the Tajogaite eruption passed under the radar of global lightning networks? (line 63)

The authors are thankful to the reviewer for asking this question. The reason why the 2021 Tajogaite eruption passed under the radar of global lightning networks is a combination of the lack of nearby sensors as well as the amplitude of the volcanic lightning not being high enough to be detected at large distances. This is why global lightning networks typically focus on major eruptions (e.g. 2018 Anak Krakatau, 2020 Taal and 2022 Hunga eruptions), as their sensors tend to have a wide geographical distribution and therefore are only able to detect very strong volcanic lightning. We have added a short explanation as suggested by the reviewer:

Lines 57-59: “However, the 2021 Tajogaite eruption passed under the radar of these networks due to the lack of nearby sensors as well as relatively low-amplitude volcanic lightning (in comparison to major eruptions), demonstrating that local electrical detectors are required^[24,25].”

3. Line 100 - this isn't clear to me, are you saying that volcanic ash plume heights did not exceed atmospheric freezing levels?

We thank the reviewer for informing us that this sentence requires some clarification. The plume height datasets of Toulouse VAAC and PEVOLCA showed that for most of the eruption volcanic ash did not exceed atmospheric freezing levels. Only on few occasions did it actually exceed the -10°C isotherm. We have rephrased the sentence to clarify this:

Lines 100-102: “The results show that volcanic ash primarily stayed below atmospheric freezing levels throughout the eruption, with an average maximum height of ~2968 m above sea level (a.s.l.) based on both datasets.”

4. Line 126 - can you comment on why the plume heights measured by the different VAAC and PEVOLCA datasets are different (they show similar trends but variation in value)

A lot of thanks to the reviewer for this question, since it can be a bit confusing. The main role of the Toulouse VAAC is to report the flight levels that are affected by volcanic ash in order to warn air traffic. Their information was based on both satellite data as well as data from the Volcano Observatory Notice for Aviation (VONA). The VONA obtained information on the plume height during different times of the day using a camera of Instituto Astrofísico de Canarias. Similarly, PEVOLCA

also used this camera, but they only reported the plume height once per day, typically in the mornings. Hence, the main difference is the time of the day when the data was obtained and the fact that the Toulouse VAAC also uses information from satellites. Both datasets have advantages and disadvantages, which is why the authors decided that it was best to show both. However, we have realized that more information is required to explain the differences between the two datasets, so the paragraph is now written as follows:

Lines 711-725: “The Toulouse Volcanic Ash Advisory Center (VAAC) reported the flight levels affected by volcanic ash based on both satellite data and data from the Volcano Observatory Notice for Aviation. The latter was compiled by the Instituto Geográfico Nacional using a camera of the Instituto Astrofísico de Canarias located 16.5 km north of the active vents at an altitude of 2365 m above sea level^[49Error! Reference source not found.]. These flight levels were converted to plume heights, providing a general trend throughout the course of the eruption^[26]. Note, however, that these values provide an upper limit, as detached ash clouds may remain at high altitudes for a long period of time even after the explosive activity at the vents has waned or stopped. Moreover, there might be a delay between the plume height and the time it is reported by the Toulouse VAAC, as information was predominantly provided at regular times during the day (03:00, 09:00, 15:00 and 21:00 UTC). We additionally included plume heights that were reported by Plan de Emergencias Volcánicas de Canarias (PEVOLCA) to provide more detail at times where the Toulouse VAAC did not report any change. These plume heights were obtained using the camera of Instituto Astrofísico de Canarias as well, but only once per day (typically mornings, but plotted here at 12:00 p.m. as a fixed time of the day)^[26].”

5. Can you give references for BTM measurements at other volcanoes (line 411)

This is a good suggestion from the reviewer. We have cited Vossen et al. (2021, 2022) as references here, which are numbers 24 and 25 on the list of references. These are the only two publications using a Biral Thunderstorm Detector to measure explosive volcanic activity so far, as this sensor is relatively new and had not been previously tested at volcanoes. The sentence now reads:

Lines 497-500: “This type of electrical activity has not been detected before by BTM measurements at other volcanoes and is distinctly different from the individual high-amplitude electrical discharges detected during strong ash emissions^[24,25].”

6. Why was the BTM moved on 26th October? (Line 477)

We thank the reviewer for this question. The BTM was moved to a closer location to have a better detection efficiency as well as for logistical reasons. We have added this short explanation to the text. At the new location, the electrical data could be telemetered to a server of INVOLCAN, which allowed us to analyse the data at various times throughout the eruption. The sentence now reads:

Lines 603-607: “The sensor was installed on 11 October 2021 at 2.65 km distance NNW from the active craters (location BTM1) and moved SW from the eruptive vents at a distance of 1.77 km (location BTM2) on 27 October 2021 to have a higher detection efficiency as a result of being closer as well as for logistical reasons, where it recorded until the end of the eruption (Figure 1).”

7. Line 459 onwards serves as a conclusion to the paper and I agree with your conclusions. Is there more you could ask for here, e.g. set out what needs to be done next to understand this further? Should BTDs be deployed at potential eruption sites to give more complete datasets?

We are grateful for these questions and particularly the interest shown here by the reviewer regarding future steps that need to be taken. To improve our understanding of the difference between the electrical signals, we will need to obtain detailed time series of the source parameters in addition to the electrical data during future eruptions. Once we understand the effect of variations in, e.g., the relative proportion of ash, the mass eruption rate and the plume height, estimates of the source parameters can be made in near real-time based on the electrical activity that is detected. Although there are many advantages of using a BTM for volcano monitoring, there are other electrical detectors that would add valuable information as well. Instead, the authors think it is more important to emphasise the value of including local electrical detectors to volcano monitoring networks. We have added a few sentences to our conclusion paragraph:

Lines 586-596: “A deeper understanding of these electrical signatures and the underlying charge mechanisms could possibly provide estimates of the relative proportion of ash, the mass eruption rate and plume height in the future. This will require further investigation during upcoming eruptions, where detailed time series of the source parameters are obtained and correlated to the electrical signatures. Moreover, our results show that local electrical monitoring of active volcanoes can provide valuable near real-time information on changes in the explosive eruption style as well as the magnitude, which may pass unobserved by regional and global lightning detection systems. The occurrence of volcanic lightning is an indicator of explosive volcanic activity without requiring the need of visibility on the crater. Hence, including electrical detectors in local monitoring networks will become increasingly more important as our ability to interpret these signals improves.”

8th May 24

Dear Dr Vossen,

Your manuscript titled "Explosive eruption style modulates volcanic electrification signals" has now been seen by our reviewers, whose comments appear below. In light of their advice we are delighted to say that we are happy, in principle, to publish a suitably revised version in Communications Earth & Environment under the open access CC BY license (Creative Commons Attribution v4.0 International License).

We therefore invite you to edit your manuscript to comply with our format requirements and to maximise the accessibility and therefore the impact of your work.

EDITORIAL REQUESTS:

*****Please take care to match our formatting and policy requirements. We will check revised manuscript and return manuscripts that do not comply. Such requests will lead to delays. *****

SUBMISSION INFORMATION:

OPEN ACCESS:

Communications Earth & Environment is a fully open access journal. Articles are made freely accessible on publication under a CC BY license (Creative Commons Attribution 4.0 International License). This license allows maximum dissemination and re-use of open access materials and is preferred by many research funding bodies.

For further information about article processing charges, open access funding, and advice and support from Nature Research, please visit <https://www.nature.com/commsenv/article-processing-charges>

At acceptance, you will be provided with instructions for completing this CC BY license on behalf of all authors. This grants us the necessary permissions to publish your paper. Additionally, you will be asked to declare that all required third party permissions have been obtained, and to provide billing

information in order to pay the article-processing charge (APC).

[link redacted]

Best regards,

Domenico Doronzo
Editorial Board Member
Communications Earth & Environment

Joe Aslin
Deputy Editor,
Communications Earth & Environment
<https://www.nature.com/commsenv/>
Twitter: @CommsEarth

REVIEWERS' COMMENTS:

Reviewer #1 (Remarks to the Author):

the authors did a great job in revising the manuscript, which is now ready for publishing. Attached an annotated manuscript with a few suggestions/corrections that can be easily addressed during the proofing stage

Reviewer #2 (Remarks to the Author):

The authors have responded to my questions on their manuscript appropriately.

[revised manuscript text omitted]

976 9419 (2022).

The reviewers' comments are given in **bold** (with line numbers corresponding to the original draft), author replies in *italics* and manuscript edits in quotation marks and regular text, with corresponding line numbers.

Reviewer 1

The authors did a great job in revising the manuscript, which is now ready for publishing. Attached an annotated manuscript with a few suggestions/corrections that can be easily addressed during the proofing stage.

We thank the reviewer for the kind words. We are very grateful to the reviewer for taking the time to go over the revised manuscript and for giving us a few more suggestions to further clarify the text.

Line 75: Replace “craters” with center or Tajogaite volcano?

We thank the reviewer for this suggestion. We agree with the reviewer that “eruptive craters” is not the best term to indicate the location of the volcanic activity in Figure 1a. We have corrected this by using the official name “Volcán de Tajogaite”. The caption of Figure 1a reads now:

Lines 74-76: “**a**) La Palma, Canary Islands (Spain), showing the location of Volcán de Tajogaite (red circle), seismic station PLPI (white triangle) and Roque de los Muchachos observatory (blue square).”

Lines 95-97: Given this mathematical relationship, did you try to normalize the electrical signals for distance³?

This is a great question from the reviewer. However, the distance does not only depend on the distance between the sensor and the active vents, but also on the location of the lightning within the plume, the height of the charged plume and the movement of the plume (thus charge) with respect to the electrical sensor. As these parameters constantly vary, the distance for which the electrical signals should be corrected is generally unknown. For this reason, the electrical signals were not corrected for the horizontal distance between the sensor and the craters. We have added a sentence that describes the different parameters that affect the amplitude of the electric field (both in the Results section pointed out by the reviewer as well as the Methods section):

Lines 97-100: “Besides the known distance between the BTM and the active vents, also the generally unknown and constantly changing height of the electrical discharges within the plume and the height and movement of the charged plume itself with respect to the BTM affect the amplitude of the electric field.”

Lines 640-642: “This distance depends on the distance between the BTM and the active vents, the height of the electrical discharges within the plume and the height and movement of the charged plume with respect to the sensor.”

Lines 155-157: Please add this brief explanation also to the main text at lines 113.

This explanation was already briefly given a few lines later (lines 123-124 in the revised manuscript), where the results regarding the normalised discharge rate are first discussed. We have only added that the normalised maximum voltage is measured per hour by the primary antenna (which is then multiplied with the electrical discharge rate to obtain the normalised discharge rate):

Lines 126-129: “The normalised discharge rate, which is the electrical discharge rate times the normalised maximum voltage measured per hour by the primary antenna (see Methods section), demonstrates that there is no relationship between these two parameters (Figures 2b-c).”

Lines 540: Add “electrical”.

We agree that clarifying that we are talking about the “electrical bursts” here is necessary to avoid confusion with the volcanic activity, which might also be considered as “bursts” by the reader. The sentence now reads:

Lines 539-543: “The duration of the electrical bursts increased from gas jetting (<30 seconds; Supplementary Figure 4e-f) to pulsating Strombolian activity (seconds to minutes) to continuous lava fountaining (predominantly minutes; Supplementary Figure 4i), which is positively correlated to a combination of relative kinetic energy, mass flux, height of the eruption column and duration of each individual explosion phase.”

Reviewer 2

The authors have responded to my questions on their manuscript appropriately.

Many thanks to the reviewer for taking the time to go over the revised manuscript.